# Requirement of Nek2a and cyclin A2 for Wapl-dependent removal of cohesin from prophase chromatin

Susanne Hellmuth [ID] [✉] & Olaf Stemmann [ID]

## Abstract

**Sister chromatid cohesion is mediated by the cohesin complex. In mitotic prophase cohesin is removed from chromosome arms in a Wapl- and phosphorylation-dependent manner. Sgo1-PP2A protects pericentromeric cohesion by dephosphorylation of cohesin and its associated Wapl antagonist sororin. However, Sgo1-PP2A relocates to inner kinetochores well before sister chromatids are separated by separase, leaving pericentromeric regions unprotected. Why deprotected cohesin is not removed by Wapl remains enigmatic. By reconstituting Wapl-dependent cohesin removal from chromatin in vitro, we discovered a requirement for Nek2a and Cdk1/2-cyclin A2. These kinases phosphorylate cohesin-bound Pds5b, thereby converting it from a sororin- to a Wapl-interactor. Replacement of endogenous Pds5b by a phosphorylation mimetic variant causes premature sister chromatid separation (PCS). Conversely, phosphorylation-resistant Pds5b impairs chromosome arm separation in prometaphase-arrested cells and suppresses PCS in the absence of Sgo1. Early mitotic degradation of Nek2a and cyclin A2 may therefore explain why only separase, but not Wapl, can trigger anaphase.**

**Keywords** Cohesin; Cyclin A2; Nek2a; Sororin; Wapl
**Subject Categories** Cell Cycle; Chromatin, Transcription & Genomics

## Introduction

The ring-shaped cohesin complex consists of four subunits, which in mitotically dividing human cells are the two structural maintenance of chromosomes (SMC) proteins Smc1α and Smc3, the kleisin Rad21 and one of two related stromal antigens, Stag1/SA1 or Stag2/SA2 (Losada et al, 1998, Losada et al, 2000). Smc1 and Smc3 each have an ABC-like ATPase head domain connected by an antiparallel, highly elongated coiled-coil to a central hinge domain that mediates Smc1-Smc3 heterodimerization (Anderson, Losada et al, 2002, Haering et al, 2002). Rad21 interacts via its N-terminal helical domain (NHD) with the head-proximal coiled-coil of Smc3

(the "neck") and via its C-terminal winged helix domain (WHD) with the Smc1 head, thereby forming a tripartite ring (Gligoris et al, 2014; Haering et al, 2004). The NHD and WHD of Rad21 are connected by a flexible linker that in its C-terminal part is associated with Stag1 or -2 and in its N-terminal part with the more loosely bound Nipbl, Pds5a or Pds5b (Hara et al, 2014, Kikuchi et al, 2016, Lee, et al, 2016, Ouyang et al, 2016). Because Stag/SA, Nipbl, and Pds5 all share extensive HEAT repeats, they are commonly referred to as HAWKs (HEAT repeat proteins associated with kleisins) (Wells et al, 2017). By topologically embracing two DNA molecules in its center, the Pds5-bound form of cohesin mediates sister chromatid cohesion in postreplicative cells (Gruber et al, 2003, Haering et al, 2008, Ochs et al, 2024). This canonical function of cohesin requires Nipbl only for the initial loading of the ring onto DNA. However, being an active ATPase, Nipbl-cohesin can extrude DNA loops in cis, thereby structuring chromatids into topologically associated domains (TADs) and influencing transcription (Davidson et al, 2019, Kim et al, 2019, Petela et al, 2018).

In human cells, the Nipbl- and ATP-dependent loading of cohesin first occurs in telophase, but this association remains dynamic throughout the G1 phase (Darwiche et al, 1999, Elbatsh et al, 2016, Gerlich et al, 2006, Watrin et al, 2006). The co-replicative establishment of sister chromatid cohesion requires the entrapment of a second DNA and is coupled with cohesion-stabilizing acetylation of the Smc3 head by Esco1/2 (Cameron et al, 2024, Hou and Zou, 2005, Murayama et al, 2018). After an exchange of Nipbl for Pds5a/b, the recruitment of sororin to acetylated cohesin competitively inhibits the binding of the anti-cohesive factor Wapl to Pds5 (Nishiyama et al, 2010). In addition, sororin is predicted to bind to the Rad21-Smc3 junction, which may contribute to its cohesion-protective function (Nasmyth et al, 2023). Cohesion allows mitotic spindle forces to be sensed and resisted, which facilitates proper amphitelic attachment of sister kinetochores. While this is critical for error-free chromosome segregation, all cohesive cohesin must be removed from chromatin before anaphase can begin. When metazoan cells enter mitosis, cohesin at chromosome arms is displaced by the action of the so-called prophase pathway (Darwiche et al, 1999, Losada et al, 1998, Waizenegger et al, 2000). This involves phosphorylation-dependent inactivation of sororin, which is replaced by Wapl as a binding partner of Pds5 (Dreier et al, 2011, Liu et al, 2013a, Nishiyama et al,

Chair of Genetics, University of Bayreuth, 95440 Bayreuth, Germany. ✉E-mail: susanne.hellmuth@uni-bayreuth.de

2013). Rad21 then dissociates from Smc3 and DNA exits the ring through this opened gate (Buheitel and Stemmann, 2013, Chan et al, 2012, Eichinger et al, 2013, Huis in 't Veld et al, 2014). Exactly how Wapl unloads cohesin is still unclear, but it may act indirectly by sequestering Rad21's N-terminus after its detachment from Smc3's neck due to ATP-driven engagement of the SMC heads (Elbatsh et al, 2016, Muir et al, 2020, Nasmyth et al, 2023). Pericentromeric cohesion is resistant to the Wapl-dependent release in mitotic prophase because it associates with its protector shugoshin 1 (Sgo1) upon phosphorylation of the latter by Cdk1-cyclin B1 (Liu et al, 2013a, McGuinness et al, 2005, Tang et al, 2004). Sgo1, in turn, recruits protein phosphatase 2A (PP2A), which maintains sororin (and cohesin) in a dephosphorylated state (Kitajima, Sakuno et al, 2006, Riedel et al, 2006, Tang et al, 2006). Pericentromeric cohesion is dissolved only at the end of metaphase, when a giant protease, separase, is activated and cleaves Rad21 (Hauf et al, 2001, Uhlmann et al, 2000). This proteolytic opening of residual cohesin triggers sister chromatid separation and marks the onset of anaphase.

The prophase pathway is important for error-free chromosome segregation and the prevention of aneuploidies (Haarhuis et al, 2013, Tedeschi et al, 2013). Next to Wapl, this proteolysis-independent opening of cohesin rings at chromosome arms requires mitotic phosphorylations. Cdk1-cyclin B1, Plk1, and aurora B have been reported as the relevant kinases, while sororin and SA2 have been identified as important substrates of phosphorylation (Dreier et al, 2011, Gimenez-Abian et al, 2004, Hauf et al, 2005, Lenart et al, 2007, Liu et al, 2013a, Losada et al, 2002, Nishiyama et al, 2013, Sumara et al, 2002). However, several issues remain. The prophase pathway has never been reconstituted in a purified system, thus leaving unanswered whether the three kinases and their two targets are sufficient for Wapl-driven release activity. Furthermore, amphitelic attachment of chromosomes to spindle microtubules coincides with the relocalization of Sgo1-PP2A from pericentromeres to inner kinetochores. The removal of Sgo1-PP2A removes a steric hindrance and allows proteolysis-promoting phosphorylation of Rad21, thereby ensuring that the subsequent cleavage of cohesin by separase is unopposed (Hauf et al, 2005, Lee et al, 2008, Liu et al, 2013b). This deprotection of cohesin and associated sororin occurs in prometaphase to metaphase, i.e., well before anaphase onset and at a time when Wapl is present and the above kinases are fully active. Why, then, does Wapl not induce premature sister chromatid separation (PCS), at least in normal mitosis? The requirement of an additional factor that is no longer present or active in metaphase may provide an answer to this question. Interestingly, two proteins, cyclin A2 and Nek2a, are known to exhibit an early mitotic degradation pattern and thus appear to be suitable to impose a temporal limit on Wapl-dependent cohesin release in mitosis.

Here we reconstitute the human prophase pathway using purified recombinant proteins. We find that the combined action of the kinases Nek2a, Cdk1/2-cyclin A2, and aurora B are necessary and sufficient for the Wapl-dependent release of cohesin and sororin from immobilized, high-salt washed G2-phase chromatin. Nek2a and Cdk1/2-cyclin A2 phosphorylate Pds5b, thereby converting it from a sororin- to a Wapl-binder. Consistently, a Pds5b variant resistant to phosphorylation by these two kinases results in the retention of cohesin on chromosome arms of prometaphase-arrested cells and suppresses premature sister

chromatid separation (PCS) in the absence of Sgo1. Conversely, a Pds5b variant that mimics constitutive phosphorylation by Nek2a and Cdk1/2-cyclin A2 causes PCS. Taken together, our study reveals Nek2a and Cdk1/2-cyclin A2 as two critical factors of prophase pathway signaling and Pds5b as an important target of these kinases. Given the early mitotic degradation of Nek2a and cyclin A2 before deprotection of pericentromeric cohesin, our study explains why anaphase is not triggered by Wapl but only by separase.

# Results

## Nek2a and Cdk1/2-cyclin A2 kinases are necessary for Wapl-dependent unloading of cohesin from isolated chromatin

Although pericentromeric cohesin is stripped from Sgo1-PP2A well before anaphase onset, Wapl cannot (and must not) trigger sister chromatid separation, at least in a healthy cell (Fig. 1A). To explain this conundrum, we hypothesized that the prophase pathway, as its name suggests, is no longer active in metaphase due to absence of required components. Nek2a and cyclin A2 are characterized by their early mitotic degradation and seem to be candidates for the proposed, limiting factors (den Elzen and Pines, 2001, Geley et al, 2001, Hames et al, 2001) (Fig. 1A). To reconstitute the prophase pathway and test this prediction, we not only tested recombinant Wapl, Cdk1-cyclin B1, Plk1, and aurora B-INCENP but included also recombinant Nek2a and Cdk1/2-cyclin A2. All five kinases were active towards model substrates and inhibited by specific inhibitors (Fig. EV1A). Wapl, Cdk1-cyclin B1, Plk1, and aurora B-INCENP were indeed not sufficient to displace cohesin and/or sororin from isolated G2 chromatin (Fig. 1B). Therefore, we included Nek2a and Cdk1/2-cyclin A2 in our release activity assays. As predicted, the addition of either kinase to the reconstitution reactions caused some cohesin and sororin, but not histone H2A, to be displaced from chromatin (Fig. 1B). Simultaneous addition of both kinases increased the release only slightly (Fig. 1B). The reconstitution assay also allowed us to assess the necessity of Plk1, aurora B-INCENP and Cdk1-cyclin B1 for the displacement of cohesin from chromatin in vitro. In the presence of Wapl, Nek2a and Cdk1/2-cyclin A2, the addition of (bacterially expressed) aurora B-INCENP resulted in the release of cohesin and sororin from chromatin, whereas the addition of Cdk1-cyclin B1 or Plk1 had no effect (Fig. 1C). Wapl, Nek2a, and Cdk1/2-cyclin A2 were purified from mitotically arrested human cells by single affinity chromatography. However, neither cyclin B1 nor Plk1 were detectable by immunoblotting in any of the three preparations (Fig. EV1B). While this confirms the dispensability of Cdk1-cyclin B1 and Plk1 for cohesin release under the conditions of our in vitro assay, it does not necessarily contradict reports of their in vivo requirement for prophase pathway signaling (see Discussion).

## Reconstitution of the prophase pathway using immobilized chromatin as substrate

To further improve our in vitro reconstitution system, we wanted to generate a biochemically more defined chromatin substrate. To this end, we used DNA-mediated chromatin pull-down (Dm-ChP)

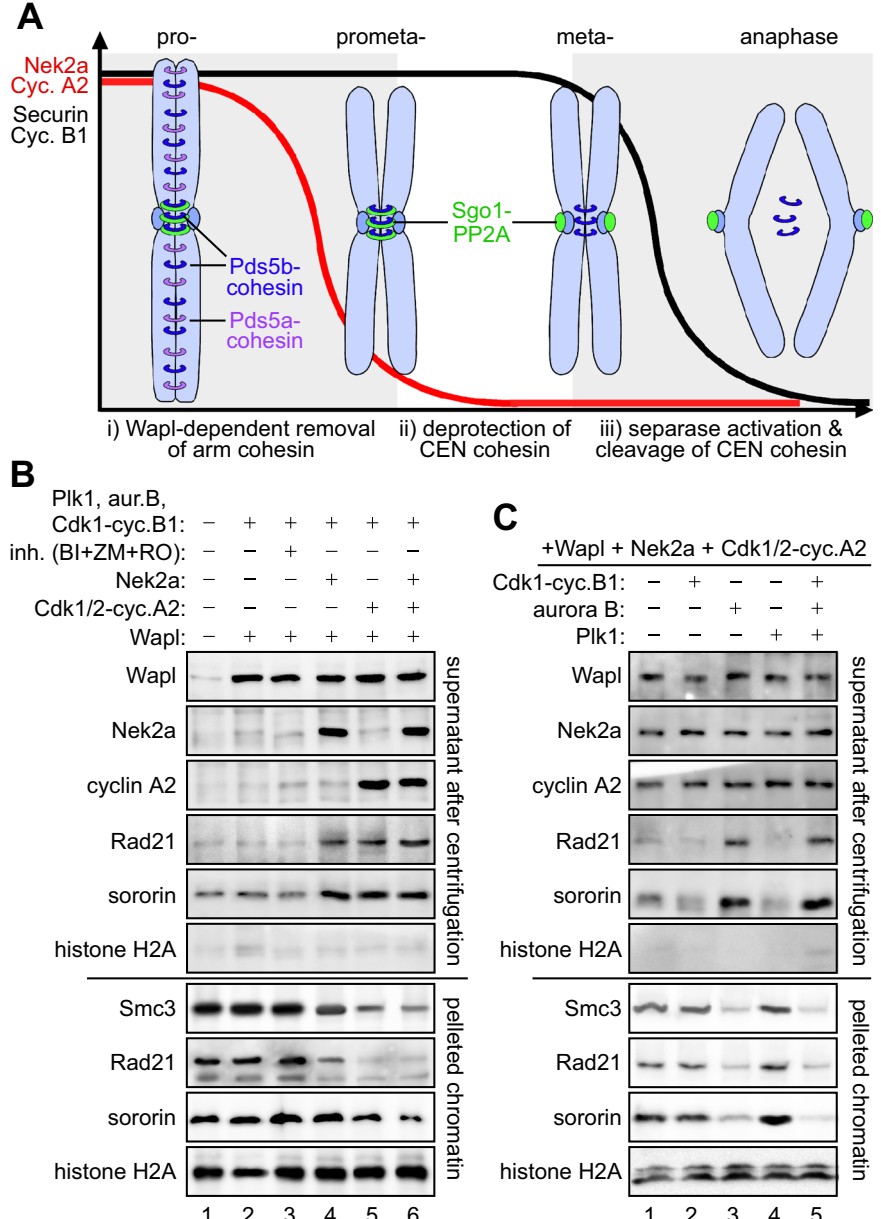

**Figure 1. Reconstitution of Wapl-dependent displacement of cohesin from crude G2-phase chromatin.**

(A) Pericentromeric cohesin is deprotected long before sister chromatids separate. During phosphorylation- and Wapl-dependent release of cohesin from chromosome arms, pericentromeric cohesin is protected by associated Sgo1-PP2A. Following the early mitotic degradation of Nek2a and cyclin A2, Sgo1-PP2A relocates to kinetochores. Yet, pericentromeric cohesion persists until securin and cyclin B1 are degraded and separase is activated. A requirement for Nek2a and cyclin A2 for prophase pathway signaling would explain why, in metaphase, Wapl can no longer release cohesin. (B) Nek2a and Cdk1/2-cyclin A2 are necessary for Wapl-dependent in vitro displacement of cohesin from chromatin. Chromatin was freshly purified by repeated sedimentation from G2-arrested HeLaK cells, incubated with recombinant Wapl, kinases, and inhibitors (BI2536, ZM-447439, and RO-3306), as indicated, and then re-pelleted. Supernatants and chromatin pellets were analyzed by immunoblotting. (C) Wapl and the three kinases Nek2a, Cdk1/2-cyclin A2, and Aurora B-INCENP are sufficient to support the displacement of cohesin from isolated chromatin. Chromatin was freshly purified by repeated sedimentation from G2-arrested HeLaK cells, incubated with recombinant Wapl, Nek2a, Cdk1/2-cyclin A2, Cdk1-cyclin B1, aurora B, and/or Plk1, as indicated, and then re-pelleted. Supernatants and chromatin pellets were analyzed by immunoblotting. Source data are available online for this figure.

technology (Aranda et al, 2019, Kliszczak et al, 2011). HeLaK cells pre-synchronized at the G1-S transition were released to undergo replication in the presence of the alkine-containing thymidine analog F-ara-EdU (Neef and Luedtke, 2011). Following G2-arrest, cell lysis, and isolation of nuclei, the DNA was covalently labeled with azide-carrying biotin by a mild "click" reaction and then immobilized on streptavidin sepharose (Fig. 2A). The corresponding beads carried native chromatin as judged by their F-ara-EdU-dependent binding of histone H2A, cohesin, Pds5b, and sororin (Fig. 2B, lanes 1–5). While histone H2A was removed by high salt,

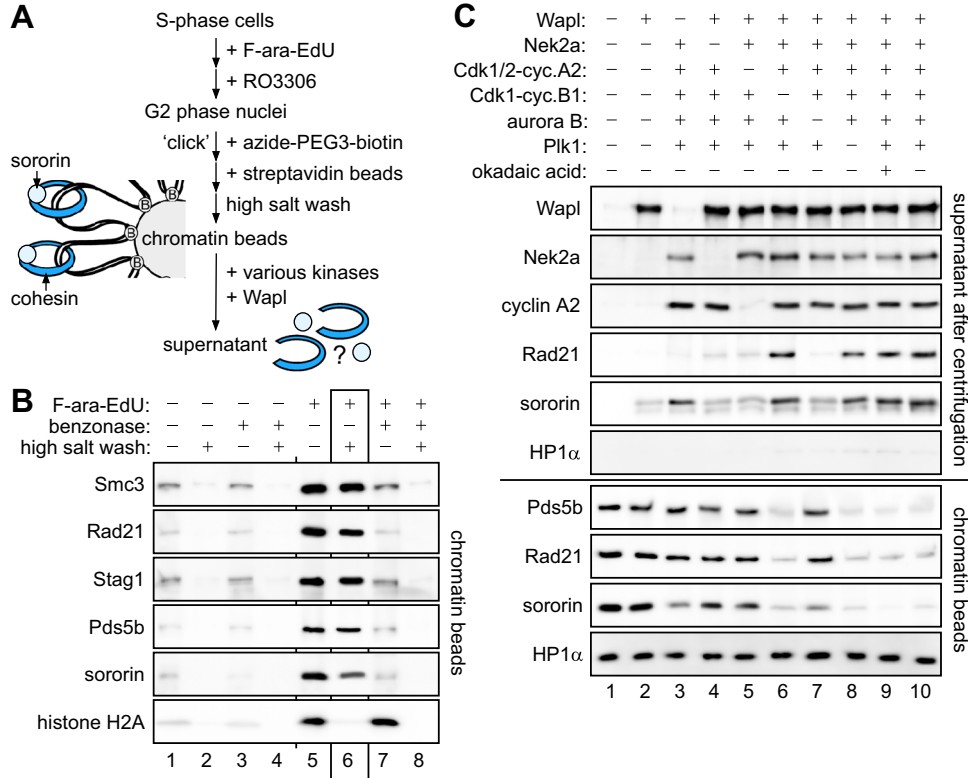

**Figure 2. Reconstitution of the prophase pathway using purified chromatin beads as substrate.**

(A) Experimental outline. DNA-mediated chromatin pull-down from G2-arrested HeLaK cells is followed by cohesin eviction experiments. (B) Cohesin is eluted from immobilized chromatin upon DNA-cleavage but not high salt treatment. Chromatin beads were treated with benzonase and/or high salt, and retained proteins were detected by immunoblotting. High-salt-washed chromatin (lane 6) was used for subsequent cohesin eviction experiments. (C) Wapl-dependent cohesin release in vitro requires Nek2a, Cdk1/2-cyclin A2, and Aurora B but not Cdk1-cyclin B1 and Plk1. Immobilized chromatin was combined with the indicated proteins. Following centrifugation, DNA-beads and supernatant were analyzed by immunoblotting. Source data are available online for this figure.

cohesin, together with Pds5b and sororin were largely retained (Fig. 2B, lane 6). Conversely and consistent with a topological entrapment of tethered DNA loops by cohesin, limited digest with benzonase stripped most cohesin, Pds5b, and sororin but not histone H2A off the beads (Fig. 2B, lane 7). Chromatin beads washed with high salt were then used as substrate in the prophase pathway reconstitution assay. Confirming our earlier experiments, the removal of Rad21, Pds5b (and sororin) from these chromatin beads required Wapl, Nek2a, Cdk1/2-cyclin A2, and aurora B-INCENP (Fig. 2C, lanes 1–5, 7, 10). In contrast and as seen before, the omission of Plk1 or Cdk1-cyclin B1 did not compromise the release of cohesin and associated factors (Fig. 2C, lanes 6 and 8). Heterochromatin protein 1 (HP1) was associated with our beads and reported to recruit Sgo1 to interphase centromeres (Kang et al, 2011). However, chemical inhibition of PP2A did not increase the efficiency of cohesin liberation (Fig. 2C, lanes 9 + 10), suggesting that the potential recruitment of PP2A by HP1-Sgo1 has no effect on our assay. Interestingly, the kinases were able to selectively solubilize some sororin in the absence of Wapl (Fig. 2C, lane 3). This is consistent with a reported phosphorylation-dependent weakening of the Pds5-sororin interaction (Dreier et al, 2011, Liu et al, 2013a, Nishiyama et al, 2013). Collectively, our reconstitution experiments confirm aurora B as a prophase pathway component. Furthermore, with Nek2a and Cdk1/2-cyclin A2, they identify two

kinases that were not previously known to be required for the proteolysis-independent release of cohesin from early mitotic chromatin.

## Nek2a and Cdk1/2-cyclin A2 phosphorylate Pds5b in its unstructured C-terminal part

To reveal putative targets of Nek2a and Cdk1/2-cyclin A2, we conducted kinase assays. Purified Nek2a phosphorylated in vitro expressed Rad21, Smc3 and Pds5b but not Smc1α or Stag1/2, whereas Cdk1/2-cyclin A2 labeled only Pds5b (Fig. EV2). We focused on Pds5b because it is specifically required for centromeric cohesion and modified by Cdk1/2-cyclin A2 on Ser1161 and −1166 in vivo (Carretero et al, 2013, Dumitru et al, 2017). In contrast to wild-type Pds5b (WT), Pds5b-Ser1161,1166Ala (2A) was resistant to phosphorylation by Cdk1/2-cyclin A2 (Fig. EV2), arguing that these two residues represent the primary target sites for this kinase within Pds5b. Inspection of nearby sequence stretches for putative Nek2a phosphorylation sites (L/M/F-x-x-S) suggested that Ser1177,−1182, and −1209 of Pds5b might be substrates (van de Kooij et al, 2019). Indeed, recombinant Nek2a modified in vitro expressed Pds5b-WT and Pds5b-2A but not Pds5b-Ser1177,1182,1209Ala (3A) (Fig. 3A, lanes 1 + 2, 11 + 12, and 15 + 16), while Cdk1/2-cyclin A2 modified Pds5B-WT and −3A

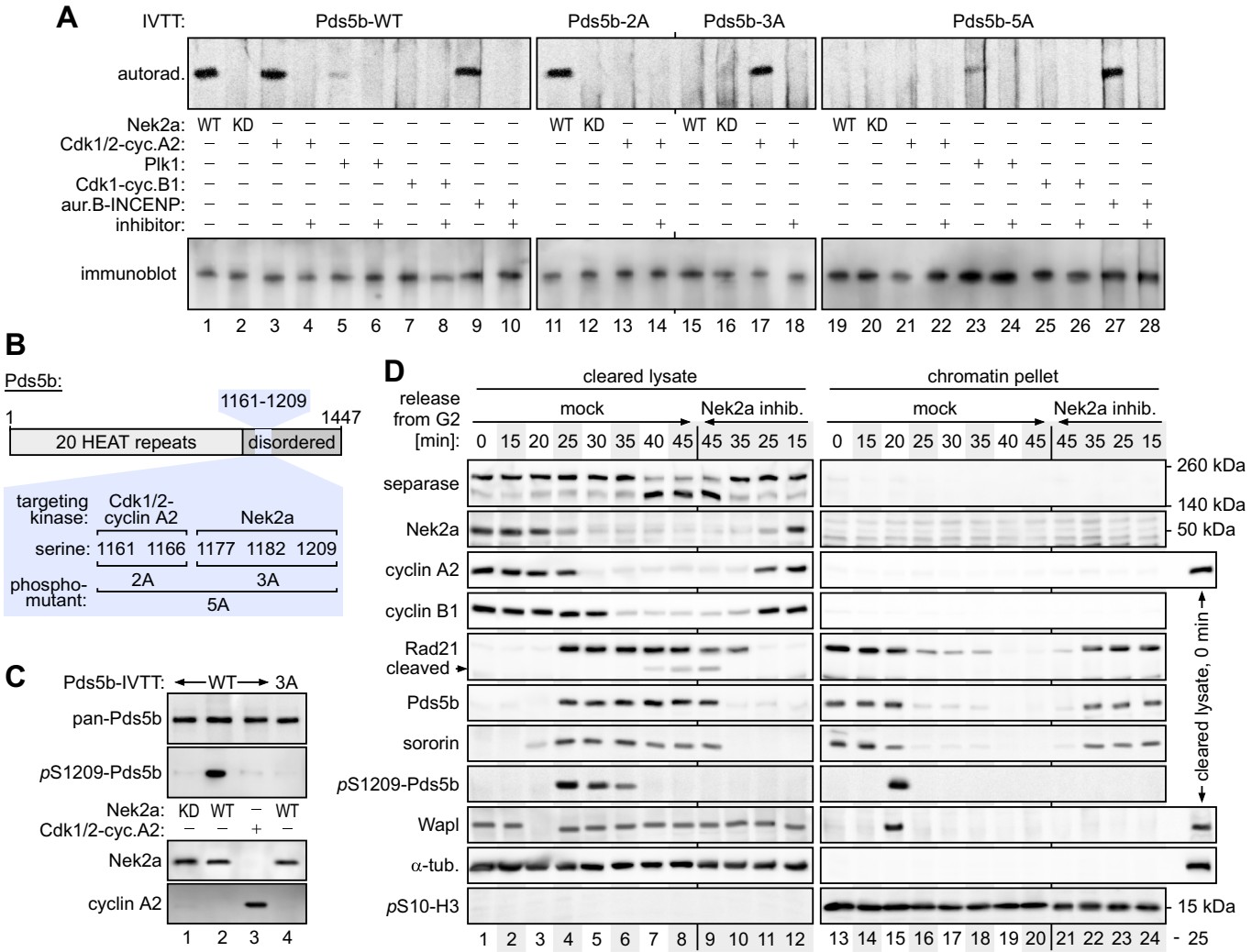

**Figure 3. Ser1209-phosphorylation by Nek2a marks the time of Wapl-dependent release of cohesin.**

(A) Nek2a and Cdk1/2-cyclin A2 target juxtaposed but distinct residues within the disordered, C-terminal domain of Pds5b. In vitro expressed Pds5b variants were incubated with the indicated kinases and inhibitors in the presence of [γ-³³P]-ATP, subjected to SDS-PAGE, and analyzed by autoradiography and immunoblotting. Note that Pds5b is also phosphorylated by aurora B and, weakly, by Plk1. KD kinase-dead Nek2a-Lys37Met; IVTT coupled in vitro transcription-translation. (B) Graphical summary of Nek2a- and Cdk1/2-cyclin A2-targeted Pds5b residues and corresponding phosphorylation-resistant variants. (C) Ser1209 of Pds5b is phosphorylated by Nek2a but not Cdk1/2-cyclin A2. Following incubation of in vitro expressed Pds5b-WT or −3A with Nek2a or Cdk1/2-cyclin A2, samples were analyzed by immunoblotting using the indicated antibodies. Nek2a-Lys37Met (KD, kinase-dead) served as a negative control. (D) The spatiotemporal dynamics of Ser1209-phosphorylation correlate with the recruitment of Wapl to chromatin followed by cohesin release. HeLaK cells synchronously cycling through mitosis were subjected to time-resolved fractionation and immunoblotting. At the time of release from a RO-3306-mediated arrest, cultures were supplemented with a Nek2a inhibitor (NCL00017509) or carrier solvent (mock). Source data are available online for this figure.

but not −2A (Fig. 3A, lanes 3 + 4, 13 + 14, and 17 + 18). Consistently, Pds5b-Ser1161,1166,1177,1182,1209Ala (5A) was resistant to phosphorylation by both kinases (Fig. 3A, lanes 19–22). In summary, recombinant Cdk1/2-cyclin A2 and Nek2a specifically target two and three serines, respectively, within a ≈50 amino acids window of Pds5b's unstructured C-terminal part (Fig. 3B).

## Phosphorylation of Ser1209 by Nek2a pinpoints Wapl-dependent cohesin release

In vivo-phosphorylation of Pds5b on Ser1177 and −1182 was previously reported (Daub et al, 2008, Olsen et al, 2010). To

investigate the phosphorylation of Ser1209, we raised a phosphorylation-specific antibody. It detected Nek2a-phosphorylated Pds5b-WT but not Cdk1/2-cyclin A2-phosphorylated Pds5b-WT or Nek2a-treated Pds5b-3A (Fig. 3C), demonstrating that S1209 is targeted by Nek2a in vitro. To follow the spatiotemporal dynamics of Ser1209-phosphorylation, HeLaK cells were synchronously released from a G2-arrest into mitosis and analyzed every 5 min by fractionation into chromatin and cytosol followed by immunoblotting (Fig. 3D). Although Wapl associates with chromosomal cohesin in interphase (Kueng et al, 2006), in G2-arrested cells with high levels of Ser-10 phosphorylated histone H3, most Wapl was already found in the soluble fraction (Fig. 3D, lanes 1 and 13). Twenty minutes after the release from the

G2-arrest, chromatin-associated Pds5b was first found phosphorylated on Ser1209. Remarkably, this correlated perfectly with the recruitment of virtually all Wapl to chromosomes (Fig. 3D, lanes 3 and 15). Five minutes later, most of Rad21, Pds5b, sororin, and Wapl had already left the chromosomes (Fig. 3D, lanes 4 and 16). Once cytosolic, Pds5b was slowly dephosphorylated, coinciding with the previous degradation of Nek2a and cyclin A2 (Fig. 3D, lanes 4–7). Chemical inhibition of Nek2a at the time of release abolished Ser1209-phosphorylation and, importantly, delayed dissociation of cohesin from chromatin until separase activation (Fig. 3D, lanes 9–12 and 21–24) (Lebraud et al, 2014). Thus, in vivo, phosphorylation of Ser1209 occurs in early mitosis and in a Nek2a-dependent manner. Furthermore, this experiment demonstrates an in vivo requirement of Nek2a for prophase pathway signaling. A variation of this experiment confirmed these results and further showed that depletion of Wapl by RNAi prevented the solubilization of Ser1209-phosphorylated Pds5b and Rad21 (Fig. EV3). In contrast, the dissociation of sororin from chromatin was less affected by the absence of Wapl but largely failed upon inhibition of Nek2a, suggesting a role for this kinase in the inactivation of sororin (Fig. EV3).

## Phosphorylation of its C-terminal part converts Pds5b from a sororin to a Wapl-binder

To study the effect of phosphorylation on Pds5 interactions, Flag-tagged Pds5a or -b were expressed in Hek293T cells either alone or together with stabilized truncation variants of Nek2a (Nek2aΔMR) and cyclin A2 (Δ86-cyclin A2) (den Elzen and Pines, 2001, Sedgwick et al, 2013). Following a taxol arrest, Pds5 was affinity-purified from chromatin-free cell lysates. The Pds5-loaded Flag-beads, which were free of endogenous Wapl and sororin, were then incubated with a mixture of recombinant Wapl and sororin before bound and unbound fractions were analyzed by immunoblotting (Fig. 4A, flow chart). Importantly, the expression of Nek2aΔMR and Δ86-cyclin A2 each improved the binding of Wapl to Pds5b at the expense of sororin. Co-expression of both switched Pds5b almost fully from a sororin- to a Wapl-specific interactor (Fig. 4A, lanes 1–4) but had no effect on the association of Pds5a with Wapl and sororin (Fig. 4A, lanes 5 and 6). In a variation of this experiment, Flag-tagged Pds5b mutants were overexpressed (Fig. 4B, lanes 1–4), affinity-purified after an extended prometaphase arrest from Nek2a- and cyclin A2-free cells and only then incubated with recombinant Nek2a and/or Cdk1/2-cyclin A2 in the presence of ATP. As before, the Flag-beads were then washed, incubated with recombinant Wapl and sororin (Fig. 4B, lane 5), washed again, and finally analyzed by immunoblotting for associated proteins (Fig. 4B, flow chart). Confirming previous kinase and competition assays, Pds5b-2A only bound sororin, even after exposure to Cdk1/2-cyclin A2, but preferred Wapl upon phosphorylation by Nek2a (Fig. 4B, lanes 6–8). Vice versa, the sororin bias of Pds5b-3A was shifted towards Wapl by Cdk1/2-cyclin A2 but not Nek2a (Fig. 4B, lanes 9–11). Importantly, Pds5b-5A constitutively bound only sororin, even after exposure to Nek2a and Cdk1/2-cyclin A2 (Fig. 4B, lanes 12–14), while Pds5b-WT treated with these two kinases was an exclusive Wapl-interactor (Fig. 4B, lane 15). Taken together, these data demonstrate that phosphorylation of Ser1161 and −1166 by Cdk1/2-cyclin A2 and of Ser1177, −1182, and −1209 by Nek2a converts Pds5b from a sororin- to a Wapl-binder. In the context of chromatin, this means Pds5b-dependent recruitment of Wapl to cohesin, thereby providing a mechanistic explanation of how these two kinases promote dissolution of arm cohesion.

## Non-degradable cyclin A2 triggers premature sister chromatid separation

A corollary of Pds5b being a target of prophase pathway signaling is that Sgo1-PP2A must be able to dephosphorylate Pds5b in order to protect pericentromeric cohesion (see next paragraph). Consequently, expression of stabilized variants of Nek2a and/or cyclin A2 might prolong the activity of the prophase pathway beyond the time of Sgo1 relocalization and, hence, trigger PCS. Although the expression of Nek2aΔMR and Δ86-cyclin A2 triggers apoptosis, this is dependent on separase activation in late metaphase (Hellmuth and Stemmann, 2020). Therefore, HeLaK cells transfected to express Nek2aΔMR and/or Δ86-cyclin A2 were taxol-arrested in prometaphase within the first cell cycle after transfection and then subjected to chromosome spreading (Fig. 5A, flow chart). Indeed, 56( ± 3)% of Δ86-cyclin A2 expressing cells suffered from PCS, while only about 3% of mock-transfected cells exhibited this defect (Fig. 5B, columns 1 and 2). Surprisingly, the expression of Nek2aΔMR had a much milder effect, causing merely 12% PCS (Fig. 5B, column 6; see Discussion). Next, we conducted rescue experiments co-expressing Pds5b variants along with the stabilized kinases. Corresponding cells were analyzed by chromosome spreads (Fig. 5B) and immunoblotting of cellular contents after their separation into soluble and chromatin fractions (Fig. 5C). The Δ86-cyclin A2 triggered PCS phenotype was largely unaffected by Pds5b-WT but almost fully suppressed by Cdk1/2-cyclin A2-resistant Pds5b-2A (Fig. 5B,C, columns/lanes 3 and 4). Additionally, this variant strongly inhibited chromosome arm separation and persisted—along with sororin—on isolated mitotic chromatin (column/lane 4). Pds5b-2A (and sororin) even persisted on zipped chromosomes in Nek2aΔMR-expressing cells (Fig. 5B,C, column/lane 8), arguing that S1161,1166-phosphorylation might be essential for Wapl-dependent cohesin removal in vivo. Conversely, Nek2a-resistant Pds5b-3A (and sororin) did not appreciably accumulate on prophase chromatin of Δ86-cyclin A2 expressing cells (Fig. 5C, lane 5). This was surprising because chemical inhibition of Nek2a prevented the release of Rad21, Pds5, and sororin from early mitotic chromatin (Figs. 3D and EV3) and indicated that constitutive S1161,1166-phosphorylation upon overexpression of Δ86-cyclin A2 might be sufficient for ring opening. Nevertheless, S1177,1182,1209-phosphorylation of Pds5b clearly contributes to prophase pathway signaling as exemplified by failure of Nek2aΔMR-expressing cells to displace Pds5b-3A (and sororin) from chromatin arms (Fig. 5B,C, column/lane 9).

## Sgo1-PP2A keeps pericentromeric Pds5b in a dephosphorylated state

Is pericentromeric Pds5b indeed kept in a dephosphorylated state? To address this issue, chromatin from prometaphase cells, which expressed Nek2aΔMR and Flag-tagged Pds5b-WT and retained cohesion primarily at pericentromeres of butterfly-shaped, condensed chromosomes (Fig. 5B,C, column/lane 7), was used as starting material for a Rad21-IP. Subsequent immunoblotting

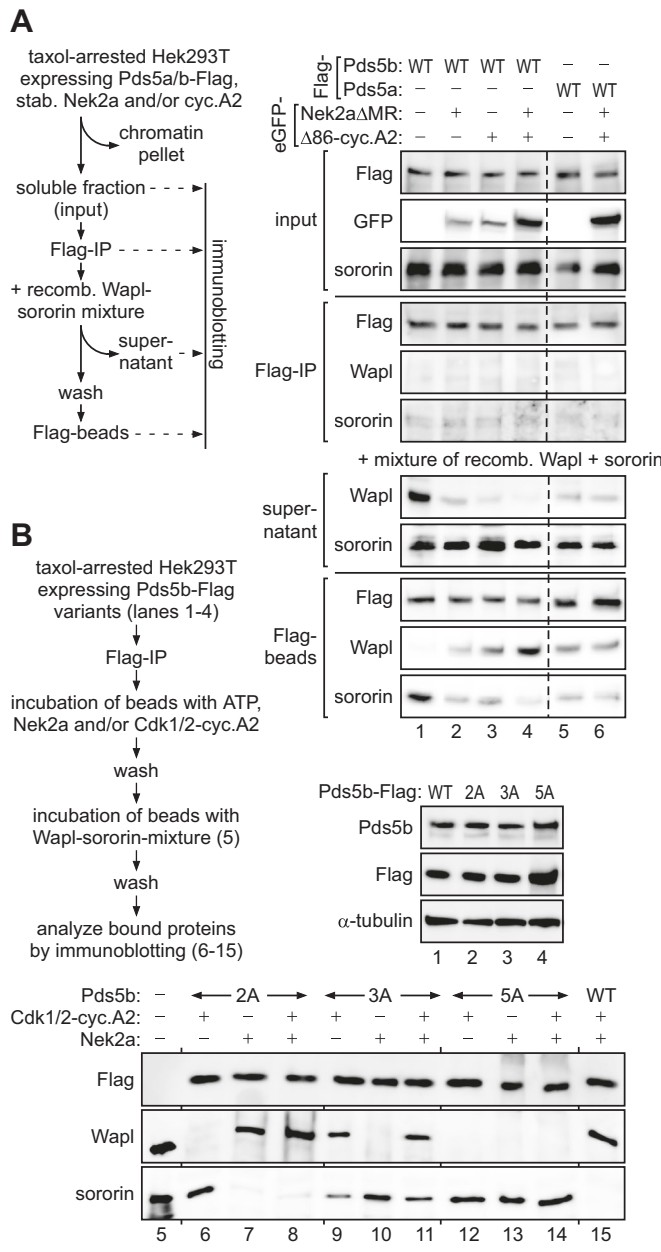

**Figure 4. Phosphorylations by Nek2a and Cdk1/2-cyclin A2 turn Pds5b from a sororin- into a Wapl-binder.**

(A) Nek2a and Cdk1/2-cyclin A2 convert Pds5b from a sororin- into a Wapl-interactor but have no effect on Pds5a's binding behavior. Shown are the experimental outline and corresponding immunoblots. Note that eGFP-Nek2aΔMR and eGFP-Δ86-cyclin A2 migrate at the same height in SDS-PAGE. (B) Rendering Pds5b resistant against both Nek2a and Cdk1/2-cyclin A2 is necessary and sufficient to switch it from a Wapl- to a sororin-binder. Shown are the experimental outline and corresponding immunoblots. Note that following the incubation with Nek2a and/or Cdk1/2-cyclin A2, ATP was no longer present to exclude phosphorylation of sororin (and/or Wapl) by traces of kinases, which might be retained on the beads. Source data are available online for this figure.

demonstrated that co-purifying endogenous and Flag-tagged Pds5b-WT were unphosphorylated at S1209 (Fig. 6A, lanes 1 and 2). However, S1209-phosphorylation became readily detectable when the Rad21-beads were treated with recombinant Nek2a (in the presence of ATP and a PP1/PP2A inhibitor) prior to Western analysis (Fig. 6A, lanes 3 and 4). Thus, in prometaphase, residual chromosomal Pds5b is unphosphorylated on S1209—even in the presence of stabilized Nek2a. These biochemical experiments were

complemented by IF analyses. (Peri)centromeres of isolated prometaphase chromosomes were stained by a pan-specific Pds5b antibody but not by anti-pS1209 (Fig. 6B, upper row). This was not due to the inability of anti-pS1209 to detect phosphorylated Pds5b in situ because it labeled mitotic chromosomes from Wapl-depleted cells along their entire length except at pericentromeres (Fig. 6B, middle rows). Importantly, this pericentromeric gap in the anti-pS1209 staining disappeared upon co-depletion of Wapl and Sgo1,

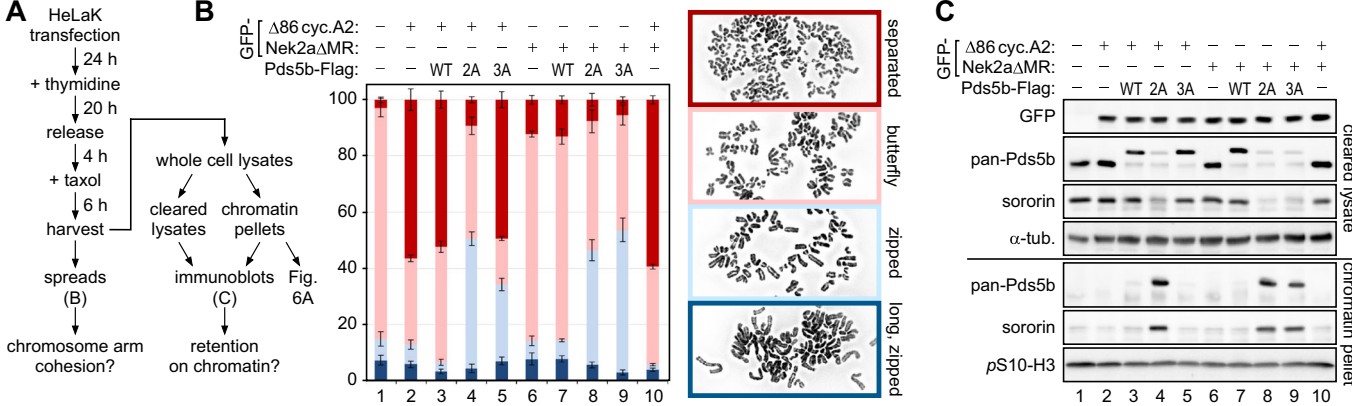

**Figure 5. Stabilized Nek2a and cyclin A2 trigger PCS to different extent.**

(A) Experimental outline. (B) Stabilized cyclin A2 causes profound PCS in prometaphase-arrested cells, which is largely suppressed by Pds5b-2A. Spreads were categorized as separated (= single chromatids), butterfly (= arms separated), zipped (= fully condensed but arms not separated), and long, zipped (early mitotic, not yet fully condensed). Note that Pds5b-2A and −3A partially prevent arm separation as judged by the relative increase of zipped chromosomes (light blue). The stacked bars represent the relative percentage (mean value with standard deviation) of each category. For every condition >100 spreads were counted for each biological replicate (n = 3). (C) Sororin is retained by Nek2a- or Cdk1/2-cyclin A2-resistant Pds5b on prometaphase chromatin. Note that (i) eGFP-Nek2aΔMR and eGFP-Δ86-cyclin A2 migrate at the same height in SDS-PAGE, (ii) expression of transgenic Pds5b results in a lower level of endogenous Pds5b, and (iii) the retention of Pds5b and sororin on chromatin correlates with the appearance of zipped chromosomes (see B). Source data are available online for this figure.

arguing that Sgo1-recruited PP2A is the relevant phosphatase (Figs. 6B, lower row and EV4). Taken together, these results strongly suggest that Pds5b at pericentromeres is kept in a dephosphorylated state by Sgo1-PP2A — at least on Nek2a-targeted Ser1209.

## Mimicking Pds5b phosphorylation causes PCS, while preventing it suppresses PCS in the absence of Sgo1

What are the cellular consequences of blocked or constitutive Pds5b phosphorylation under more physiological conditions, i.e., without expression of APC/C-resistant Nek2a or cyclin A2? To address this issue, HeLaK cells were transfected to deplete Pds5a/b by RNAi and simultaneously express siRNA-resistant, Flag-tagged Pds5b-variants. In an attempt to mimic constitutive phosphorylation, a new Pds5b-6D (Ser1161,1162,1166,1177,1182, and 1209Asp) was also included. Corresponding cells were synchronized in prometaphase and subjected to Flag-IP and chromosome spreading (Fig. 7). Wild-type Pds5b-Flag robustly interacted with Wapl but sororin-binding was undetectable (Fig. 7A, lane 1). While Pds5b-6D behaved like wildtype (Fig. 7A, lane 5), the picture was inverted for Pds5b-2A, −3A, and −5A. These Nek2a- and/or Cdk1/2-cyclin A2-resistant variants bound little Wapl but associated with sororin instead (Fig. 7A, lanes 2–4). This exchange of Wapl for sororin, which was most pronounced for Pds5b-5A, correlated with increasing chromatin association as judged by co-IP of histone H2A. Chromosome spreading revealed that most control cells (73%) contained cohesed chromosomes with separated arms and that only a few (3%) suffered from PCS (Fig. 7B, column 1). Replacing Pds5b-WT with Pds5b-2A, -3A, or -5A stepwise increased the fraction of cells containing chromosomes with unseparated/zipped arms from 24% (WT) to 74% (5A) (Fig. 7B, columns 2–4). In contrast, 51% of cells expressing Pds5b-6D contained single chromatids (Fig. 7B, column 5). Depletion of

endogenous Sgo1 by RNAi did not visibly change the interaction behavior of the Pds5b variants (Fig. 7A, lanes 6–10), which is consistent with Sgo1-PP2A protecting only a small pool of cohesin. However, and consistent with previous reports (McGuinness et al, 2005, Salic et al, 2004, Tang et al, 2004), absence of Sgo1 caused PCS in 63% of Pds5b-WT expressing cells (Fig. 7B, column 10). While Pds5b-6D further aggravated this phenotype (PCS in 75% of cells, Fig. 7B, column 6), Ala-containing Pds5b variants increasingly suppressed it, with Pds5b-5A having the strongest rescue effect (PCS in 12% of cells, Fig. 7B, column 7). Thus, preventing phosphorylation of Pds5b by Nek2a and Cdk1/2-cyclin A2 renders cohesin partially resistant to Wapl-dependent removal from prophase chromatin. Conversely, constitutive phosphorylation of Pds5b by Nek2a and Cdk1/2-cyclin A2 can be successfully mimicked by Asp-residues and renders the cohesin ring hypersensitive to opening in early mitosis. Mechanistically, these effects can be attributed to Pds5b-5A and -6D binding sororin constitutively or not at all, respectively.

## A kinase-independent ATP requirement for cohesin release

Based on AlphaFold predictions, ATP-driven engagement of Smc1/3 heads may result in a distortion of Smc3's neck followed by dissociation of Rad21's NHD. Wapl may then sequester Rad21's N-terminus, thereby keeping the cohesin ring open long enough for DNA to slip out (Nasmyth et al, 2023). To probe for a putative ATP requirement during ring opening, we slightly modified our release assay and sequentially incubated the chromatin beads first with kinases (and ATP) and only after intermediate washes with Wapl in the presence or absence of ATP (Fig. 8A). At the same time, we asked whether Pds5a and Nipbl, which were also associated with our immobilized chromatin, would behave like Pds5b and also be displaced under the conditions of our release assay. Indeed, Pds5a

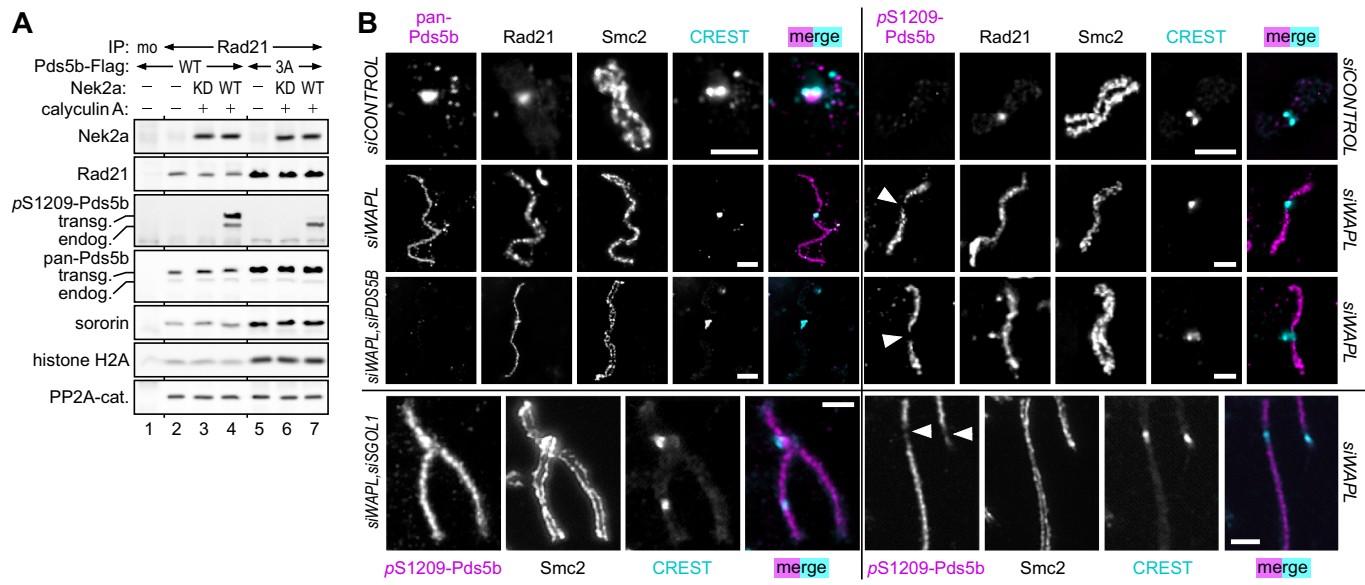

**Figure 6. Sgo1-PP2A keeps pericentromeric Pds5b in a dephosphorylated state.**

(A) Chromatin from Nek2aΔMR- and Flag-tagged Pds5b-WT-expressing cells (see column/lane 7 of Fig. 5B, C) was benzonase-treated and subjected to Rad21-IP. Precipitated proteins were detected by immunoblotting. Chromatin from Nek2aΔMR- and Flag-tagged Pds5b-3A-expressing cells (see column/lane 9 of Fig. 5B, C) served as control. mo mock-IP; KD kinase-dead; transg transgenic; endog endogenous. (B) HeLaK cells were transfected with the indicated siRNAs (see Fig. EV4) and released from G2- into a prometaphase arrest. Chromosomes were isolated and subjected to immunofluorescence microscopy using the indicated antibodies. Arrowheads point at pericentromeric regions as judged by CREST staining. Scale bars = 2.5 μm. Source data are available online for this figure.

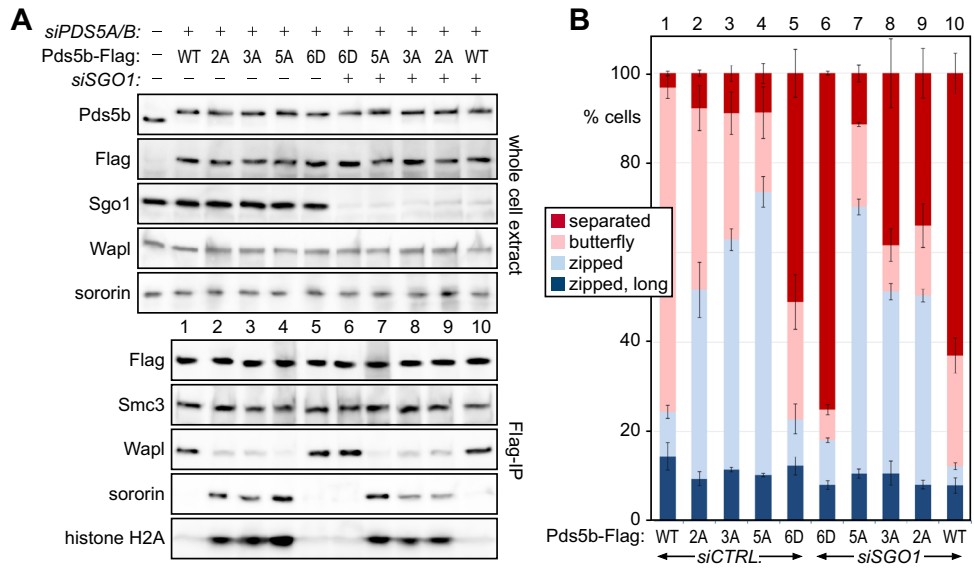

**Figure 7. Mimicking Pds5b phosphorylation by Nek2a and Cdk1/2-cyclin A2 causes PCS, while preventing it inhibits arm separation and suppresses PCS in the absence of Sgo1.**

(A) HeLaK cells were transfected to replace endogenous Pds5a/b by Flag-tagged Pds5b variants and, where indicated, to deplete Sgo1. Following synchronization in prometaphase, cells were analyzed by (IP-)Western. (B) Chromosome spreads from cells in A were analyzed as in Fig. 5B. Note that the co-IP of histone H2A with Pds5b correlates with the abundance of zipped chromosomes, which is consistent with chromosomal retention of cohesin. The stacked bars represent the relative percentage (mean value with standard deviation) of each category of chromosome morphology. For every condition >100 spreads were counted for each biological replicate (n = 3). Source data are available online for this figure.

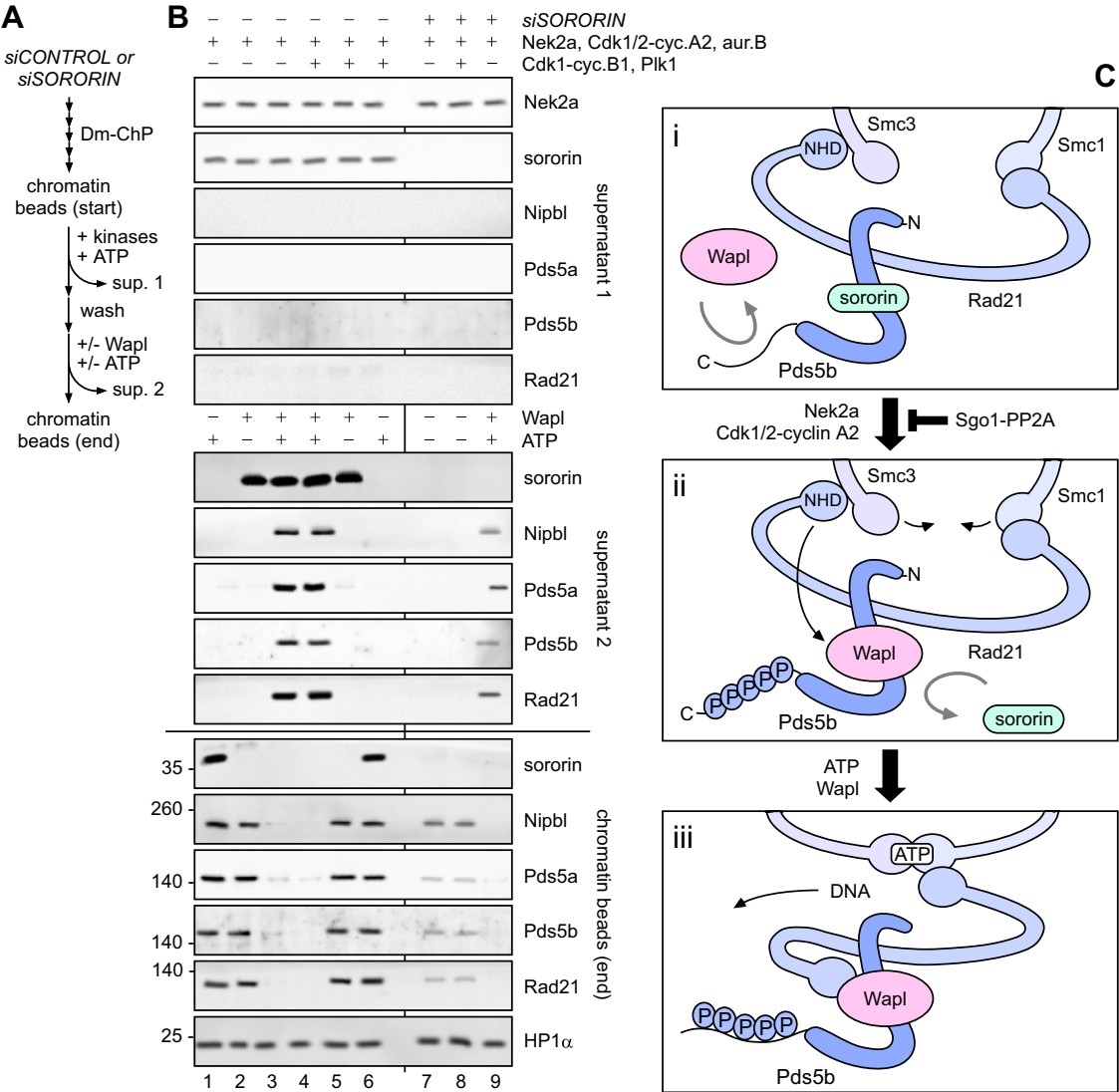

**Figure 8. A phosphorylation-independent ATP requirement for Wapl-dependent release of cohesin.**

(A) Experimental outline. See Fig. 2A for the steps of DNA-mediated chromatin pull-down (Dm-ChP). Sup supernatant. (B) In vitro Pds5a, Pds5b, and Nipbl are all susceptible to Wapl-dependent release. Immobilized G2 chromatin from sororin-depleted or mock-treated cells was incubated with the indicated kinases, washed, and then exposed to Wapl and/or ATP. Supernatants and remainder chromatin were analyzed by immunoblotting using the indicated antibodies. Note that the liberation of Pds5a, Pds5b, and Nipbl, but not sororin, requires ATP in addition to Wapl. (C) Model. In mitotic prophase, Nek2a and Cdk1/2-cyclin A2 phosphorylate Pds5b within its unstructured C-terminal domain, thereby converting it from a sororin- to a Wapl-binder (i and ii). Sgo1-PP2A-dependent dephosphorylations preserve the Pds5b-sororin interaction at pericentromeres. Work from others suggest that ATP-dependent engagement of SMC heads then result in transient detachment of Rad21's NHD from Smc3's neck (Nasmyth et al, 2023). The same authors further proposed that sequestration of Rad21's N-terminus by Wapl stabilizes the open state of the gate, thereby giving DNA time to exit the cohesin ring (ii and iii). Note that DNA, SA1/2, and additional phosphorylations were omitted for the sake of clarity. Source data are available online for this figure.

and Nipbl behaved like Pds5b (and Rad21) and eluted upon exposure to Nek2a, Cdk1/2-cyclin A2, aurora B-INCENP, Wapl, and ATP, while Cdk1-cyclin B1 and Plk1 were again dispensable (Fig. 8B, lanes 1–6). Importantly, the release of all cohesin subunits failed when ATP was omitted during the incubation of pre-phosphorylated chromatin beads with Wapl (Fig. 8B, lanes 2 and 5). This was in marked contrast to the behavior of sororin, which was displaced by Wapl even in the absence of ATP. We conclude that there is a kinase-independent ATP requirement for the release of cohesin in the presence of Wapl. Our data further suggest that

Pds5a-, Pds5b- and Nipbl-bound forms of cohesin are all susceptible to Wapl-dependent release, at least under the conditions of our assay (see discussion). Finally, to test whether phosphorylation might be sufficient to displace sororin-free cohesin from DNA, we also prepared chromatin beads from sororin-depleted G2-phase cells. Not surprisingly, this resulted in beads with a much-reduced cohesin load. Nevertheless, Pds5a/b, Nipbl, and Rad21 were detectable, did not elute upon incubation with Nek2a, Cdk1/2-cyclin A2, and aurora B-INCENP alone but still required Wapl for their release into the supernatant (Fig. 8B,

lanes 7–9). Taken together, our results are consistent with cohesin release occurring in two steps, a phosphorylation-driven competitive displacement of sororin by Wapl followed by Wapl- and ATP-dependent opening of the ring (Fig. 8C).

## Discussion

Here, using purified components, we report the successful reconstitution of the non-proteolytic displacement of cohesin from replicated chromatin. Next to Wapl and ATP, the cohesin release in our system required only three kinases, namely aurora B, Nek2a, and cyclin A2-activated Cdk1/2, whereas, surprisingly, Plk1 and Cdk1-cyclin B1 were dispensable. The requirement for Nek2a and cyclin A2 strongly suggests that the Wapl-dependent removal of cohesin from chromatin is no longer active by metaphase, when these two factors have already been degraded. This explains why, in normal mitosis, there is no Wapl-dependent PCS in the time between deprotection of pericentromeric cohesin (by the departure of Sgo1-PP2A) until the activation of separase.

The dispensability of Plk1 and Cdk1-cyclin B1 for the in vitro release of cohesin from chromatin is surprising because several independent studies clearly showed the requirement of these two kinases for prophase pathway signaling in cells (Dreier et al, 2011, Gimenez-Abian et al, 2004, Lenart et al, 2007, Liu et al, 2013a, Losada et al, 2002, Nishiyama et al, 2013, Sumara et al, 2002). We cannot exclude that sites important for cohesin release are phosphorylated by recombinant Nek2a, Cdk1/2-cyclin A2, and/or aurora B in vitro but actually targeted by Plk1 and/or Cdk1-cyclin B1 in vivo. Alternatively, the dispensability of Plk1 and Cdk1-cyclin B1 could be explained if prophase pathway components are activated by these kinases in vivo, which are added in their already active, phosphorylated form to our in vitro system. In this context, it is interesting to note that Wapl was reported as a putatively Plk1-activated prophase pathway component (Challa et al, 2019). Notably, we isolate Wapl from mitotically arrested cells, i.e. at a state of high Plk1 and Cdk1-cyclin B1 activity. Therefore, it will be interesting to test whether phosphatase treatment of Wapl may be sufficient to establish a Plk1 and/or Cdk1-cyclin B1 requirement for cohesin release in our assay. Finally, given their similarities, the actual in vivo requirement of Cdk1-cyclin B1 and/or Cdk1/2-cyclin A2 for prophase pathway signaling needs careful evaluation. While both kinases clearly have similar substrate specificities (Moore et al, 2003), in our hands, Cdk1/2-cyclin A2 can phosphorylate Pds5b, while Cdk1-cyclin B1 cannot (Fig. 3A). Thus, it is conceivable that Cdk1/2-cyclin A2 can functionally replace Cdk1-cyclin B1 in prophase pathway signaling, but not vice versa.

Cyclin A2-resistant Pds5b-2A (and sororin) accumulated on mitotic chromatin - even in cells expressing Nek2aΔMR (Fig. 5C). This suggests that phosphorylation of Pds5b on serines 1161 and 1166 may be essential for Wapl-dependent cohesin removal and cannot be functionally replaced by Nek2a-dependent phosphorylation of nearby sites. Conversely, the accumulation of Nek2a-resistant Pds5b-3A (and sororin) on mitotic chromatin could be prevented by co-expression of Δ86-cyclin A2 (Fig. 5C), suggesting that continued phosphorylation of Pds5b on S1161,1166 may be sufficient to compensate for the loss of Nek2a. These interpretations may also explain why Δ86-cyclin A2 expression caused

profound PCS, whereas the premature loss of cohesion caused by Nek2aΔMR expression was almost negligible (Fig. 5B). Thus, only the stabilization of cyclin A2, but not of Nek2a, seems to prolong the activity of the prophase pathway beyond the time of cohesin deprotection by departure of Sgo1-PP2A. Nevertheless, under physiological conditions, i.e., without overexpression of Δ86-cyclin A2, Nek2a is clearly important for prophase pathway signaling. This is exemplified by the profound retention of Rad21, Pds5b, and sororin upon chemical inhibition of Nek2a (Figs. 3D and EV3) and the ability of Nek2a-resistant Pds5b-3A to partially suppress PCS caused by siRNA-mediated depletion of Sgo1 (Fig. 7B).

Nek2a- and Cdk1/2-cyclin A2 phosphorylate Pds5b at 5 serines located within a 50 amino acids window of the long (330 residues), unstructured C-terminal part. How exactly these phosphorylations convert Pds5b from a sororin to a Wapl-binder, we cannot yet say. However, it is interesting to note that Nek2a, Cdk1/2-cyclin A2, and aurora B are able to selectively elute some sororin (but no cohesin) from immobilized chromatin in the absence of Wapl (Figs. 2C and 8B). This suggests that the primary effect of phosphorylation is the weakening of sororin's interaction with Pds5-cohesin rather than the enforcement of Wapl's interaction with these partners. With their shared YSR- and FGF-motifs, sororin, and Wapl compete for the same contact sites on Pds5, but AlphaFold predicts an additional interaction of sororin with the Smc3-Rad21 interface, which is specific, i.e., not shared by Wapl (Nasmyth et al, 2023). As it is difficult to envision how phosphorylation of Pds5b's C-terminal part could have different effects on very similar interactions, we favor the idea that it may instead interfere with the interaction of sororin with cohesin's DNA exit gate.

The phosphorylation sites within Pds5b targeted by Nek2a- and Cdk1/2-cyclin A2 are not conserved in Pds5a. The serines are either absent (positions 1161, 1177, and 1209) or in an altered sequence context that no longer matches the substrate consensus sequences of the two kinases (positions 1166 and 1182). Accordingly, pre-exposure of Pds5a to Nek2a and Cdk1/2-cyclin A2 does not modulate its subsequent binding to sororin or Wapl (Fig. 4A). Nevertheless, Pds5a-cohesin is clearly subject to Wapl-dependent release; otherwise spread metaphase chromosomes would not display separated arms. Consistently, Pds5a is readily released from chromatin in presence of Nek2a, Cdk1/2-cyclin A2, aurora B, Wapl and ATP (Fig. 8B). Since Aurora B-dependent phosphorylations appear to be insufficient for cohesin release (Fig. 1B), Nek2a and/or Cdk1/2-cyclin A2 are likely required also for opening of Pds5a-cohesin. This suggests the existence of an additional target of these kinases, with sororin being an obvious candidate. Alternatively, Rad21 and/or Smc3, both of which are phosphorylated by Nek2a in vitro (Fig. EV2), may be relevant substrates. This is because their NHD and neck region, respectively, represent cohesin's exit gate, which is predicted to also make contact with sororin (Nasmyth et al, 2023).

We also observed mobilization of Nipbl in our assay (Fig. 8B). This is somewhat surprising because the Wapl-dependent release of Nipbl-cohesin is believed to depend on the prior exchange of Nipbl for Pds5 at CTCF-sites (Nasmyth et al, 2023, Wutz, Varnai et al, 2017). However, rules might be different in mitosis (and the mitosis-like state that Nek2a, Cdk1/2-cyclin A2, and aurora B will likely create in our assay). It is conceivable that a pool of cohesin

still actively extrudes chromatin loops at mitotic entry. Yet, most, if not all, cohesin is removed from chromosome arms in prophase, and this is important for the reprogramming of gene expression in the next cell cycle (Perea-Resa et al, 2020). Therefore, it seems possible that mitotic phosphorylations enable the Wapl-dependent release of Nipbl-cohesin, for example, by rendering the replacement of Nipbl by Pds5 independent of CTCF or by facilitating a direct Wapl-Nipbl interaction.

While the exact mechanism of Wapl-dependent cohesin release is still unknown, an interesting model was recently proposed based on AlphaFold predictions (Nasmyth et al, 2023). According to this model, the ATP-dependent engagement of the SMC1/3 head domains results in the initial opening of the Smc3-Rad21 gate. Subsequent binding of Wapl's C-terminal domain to the 35 most N-terminal amino acids of Rad21 prevents closure and keeps the gate open long enough for DNA to exit the ring (Nasmyth et al, 2023). Our biochemical release assay provides the first experimental support for the proposed ATP requirement (Fig. 8B). Surprisingly, exposure of cohesin to ATP prior to the addition of Wapl (i.e., during the pre-phosphorylation step) was not sufficient for release. This suggests that as long as Wapl is still missing, the head engagement is transient and possibly reversed upon re-binding of Rad21 to Smc3. Our reconstitution system should also be ideal for assessing the importance of the proposed sequestration of Rad21's N-terminus by Wapl. For example, it will be interesting to test whether an antibody directed against a Rad21 N-terminal peptide can functionally replace Wapl in cohesin release.

# Methods

### Reagents and tools table

| Reagent/ Resource | Reference or source | Identifier or catalog number |
|---|---|---|
| **Experimental Models** | | |
| HeLaK (K=Kyoto) | Gift from T. U. Mayer | RRID: CVCL_1922 |
| Hek293T | Gift from M. Kirschner | RRID: CVCL_0063 |
| *E. coli.* Rosetta 2 DE3 | Novagen | 71397 |
| **Recombinant DNA** | | |
| pCS2-Pds5a/b-FLAG$_3$-Tev$_2$ | This study pCS2 backbone: Gift from M. Kirschner; insert: PCR on human cDNA (selfmade) | N/A |
| pCS2-GFP-SUMOStar-Wapl | This study pCS2 backbone: Gift from M. Kirschner; insert: PCR on human cDNA (selfmade) | N/A |
| pCS2-GFP-SUMOStar-Nek2aΔMR/Δ86-cyclin A2 | Hellmuth and Stemmann, 2020, https://doi.org/10.1038/s41586-020-2187-y. | N/A |

| Reagent/ Resource | Reference or source | Identifier or catalog number |
|---|---|---|
| **Antibodies** | | |
| Smc1 Rabbit pAb (WB dilution 1:1,000) | Bethyl | A300-055A |
| Smc3 Rabbit pAb (WB dilution 1 μg/ml) | Schöckel, et al, 2011, https://doi.org/10.1038/ncb2280. | N/A |
| Rad21 Guinea Pig pAb (WB dilution 1.8 μg/ml) | This study. Antigen: A107-S271 of the human protein | N/A |
| Rad21 (B2) Mouse mAb (WB dilution 1:800) | Santa Cruz Biotechnology | Sc-271601 |
| Rad21 Rabbit pAb (WB dilution 1:1,000) | Bethyl | A3000-080A |
| Rad21 Mouse mAb (IF dilution 1:500) | Millipore | 05-908 |
| Smc2 Rabbit pAb (IF dilution 2 μg/ml) | Hellmuth et al, 2020, https://doi.org/10.1038/s41586-020-2182-3. | N/A |
| Stag1 Rabbit pAb (SA1; WB dilution 3.5 μg/ml) | Kindly provided by Susannah Rankin, University of Oklahoma, USA. | N/A |
| Stag2 Rabbit pAb (SA2; WB dilution 3.3 μg/ml) | Kindly provided by Jan-Michael Peters, Institute of Molecular Pathology, Austria | N/A |
| *p*S1209-Pds5b Guinea Pig pAb (WB dilution 0.25 μg/ml; IF dilution 1.5 μg/ml) | This study. Antigen: CDLVR*p*SELEK (Cys+ amino acids 1205-1213 of the human protein) | N/A |
| Pan-Pds5b Guinea Pig pAb (WB and IF dilution 1.5 μg/ml) | This study. Antigen: P1137-M1308 of the human protein | N/A |
| Pds5a Rabbit pAb (WB dilution 1:1,000) | Proteintech | 17485-1-AP |
| Nipbl (C-9) Mouse mAb (WB dilution 1:800) | Santa Cruz Biotechnology | Sc-374625 |
| Cyclin A2 Mouse mAb (WB dilution 1:1,000) | Santa Cruz Biotechnology | 46B11 |
| Nek2 (clone 20) Mouse mAb (WB dilution 1:800) | BD Transduction Laboratories | 610549 |
| Sororin Mouse mAb (WB dilution 1:1,000) | Santa Cruz Biotechnology | Sc-365319 |

     

| Reagent/ Resource | Reference or source | Identifier or catalog number |
|---|---|---|
| Sororin Rabbit pAb (WB dilution 1.7 μg/ml) | Wolf et al, 2018, https://doi.org/ 10.1242/ jcs.212100 | N/A |
| Wapl Guinea Pig pAb (WB dilution 1.2 μg/ml) | This study. Antigen: M1-C625 of the human protein | N/A |
| Sgo1 Rabbit pAb (WB dilution 1:500) | Abcam | Ab21633 |
| Sgo1 Guinea Pig pAb (WB dilution 2.5 μg/ml) | This study. Antigen: S105-K281 of the human isoform 1 | N/A |
| PP2A-C (1D6) Mouse mAb (WB dilution 1:1,000) | Millipore | 05-421 |
| HP1-alpha Mouse mAb (hybridoma supernatant; WB dilution 1:1,000) | DSHB, University of Iowa, USA | PCRP-CBX5-2D8 |
| HP1-alpha (F.747.6) Mouse mAb (WB dilution 1:1,000) | Invitrogen | MA5-14989 |
| H2A Rabbit pAb (WB dilution 1:1,000) | Abeomics | 11-7017 |
| pS10-histone H3 Rabbit pAb (WB diluton 1:1,000) | Merck | 06-570 |
| alpha-Tubulin (12G10) Mouse mAb (WB dilution 1:200) | DSHB, University of Iowa, USA | N/A |
| Flag (M2) Mouse mAb (WB dilution 1:2,000) | Sigma-Aldrich | F1804 |
| GFP (clone 71) Mouse mAb (hybridoma supernatant; WB dilution 1:2,000) | Gift from D. van Essen and S. Saccani | N/A |
| CREST Human pAb (IF dilution 1:1,000) | ImmunoVision | Hct-0100 |
| Goat anti-mouse IgG (H/L):HRP | Sigma-Aldrich | 12-349 |
| Goat anti-rabbit IgG (H/L):HRP | Sigma-Aldrich | 12-348 |
| Goat anti-guinea pig IgG (H/L):HRP | Sigma-Aldrich | AP108P |
| Goat anti-guinea pig IgG (H/L):Alexa Fluor™ 546 | Invitrogen | A-11074 |

| Reagent/ Resource | Reference or source | Identifier or catalog number |
|---|---|---|
| Donkey anti-rabbit IgG (H/L):Alexa Fluor™ 546 | Invitrogen | A10040 |
| Goat anti-mouse IgG (H/L):Alexa Fluor™ 488 | Invitrogen | A-11001 |
| Goat anti-mouse IgG (H/L):Alexa Fluor™ 350 | Invitrogen | A-11045 |
| Goat anti-rabbit IgG (H/ L):DyLight™ 488 | Bethyl | A120-101D2 |
| Goat anti-human IgG (H/ L):DyLight™ Cy5 | Bethyl | A80-304D3 |
| **Oligonucleotides and other sequence-based reagents** | | |
| Human PDS5B siRNA | This study. | 5'-GGAUAGAUCUUAAGCAGUATT-3' |
| Human PDS5A_1 siRNA | This study. | 5'-GGGAAAGAACACUGGAUAATT-3' |
| Human PDS5A_2 siRNA | This study. | 5'-UGUAAAAGCUCUCAACGAATT-3' |
| Human WAPL_1 siRNA | This study. | 5'-CGGACUACCCUUAGCACAA-3' |
| Human WAPL_2 siRNA | This study. | 5'-GGUUAAGUGUUCCUCUUAUUTT-3' |
| Human SGO1_1 siRNA | This study. | 5'-AGUAGAACCUGCUCAGAA-3' |
| Human SGO1_2 siRNA | This study. | 5'-GAUGACAGCUCCAGAAAUUTT-3' |
| Human SORORIN_1 siRNA | This study. | 5'-AUUGUACCUUUUGAUGUUUAGAAGU-3' |
| Human SORORIN_2 siRNA | This study. | 5'-CCCCUUUAGUUCUGUAAAUAGUCCC-3' |
| **Chemicals, enzymes, and other reagents** | | |
| Lipofectamine 2000 | Invitrogen | 11668019 |
| Lipofectamine RNAiMAX | Invitrogen | 13778150 |
| DMEM | Biowest | L0102 |
| FCS | Sigma-Aldrich | F4135 |
| Thymidine | Sigma-Aldrich | T1895 |
| Taxol | LC-Laboratories | P-9600 |
| RO-3306 | Santa Cruz Biotechnology | sc-358700A |
| Okadaic acid | LC-Laboratories | O-5857 |
| BI2536 | MCE | HY-50698 |
| ZM-447439 | Tocris | 2458 |
| Staurosporine | Abcam | ab120056 |

| Reagent/ Resource | Reference or source | Identifier or catalog number |
|---|---|---|
| Imject™ Maleimide-Activated mcKLH | Thermo Fisher Scientific | 10720995 |
| NHS-activated Sepharose 4 Fast Flow | GE-Healthcare/ Cytiva | GE17-0906-01 |
| SulfoLink™ coupling resin | Thermo Fisher Scientific | 20401 |
| cOmplete™ Protease Inhibitor cocktail EDTA-free | Roche | 11836170001 |
| Calyculin A | LC-Laboratories | C-3978 |
| Benzonase nuclease | Santa Cruz Biotechnology | Sc-202391 |
| rProtein A Sepharose™ Fast Flow | GE-Healthcare/ Cytiva | GE17-1279-01 |
| (2'S)-2'-Deoxy-2'-fluoro-5-ethynyluridine, (F-ara-EdU) | Sigma-Aldrich | T511293 |
| (+)-Sodium-L-ascorbate | Sigma-Aldrich | A7631 |
| Azide-PEG3-biotin | Sigma-Aldrich | 762024 |
| Cooper(II)sulfate | Roth | P023.1 |
| Pierce™ Streptavidin-Agarose high capacity | Thermo Fisher Scientific | 20357 |
| Ni-NTA agarose | Qiagen | 30210 |
| Recombinant human active PLK1 Protein | R&D Systems | 3804-KS-010 |
| Recombinant human histone H1 | NEB | M2501 |
| Recombinant MBP | Upstate®/Merck | 13-173 |
| TᴺT® Quick Coupled Transcription/ Translation System (SP6) | Promega | L-2080 |
| Fluorobind PVDF membrane | Serva | 42571.01 |
| **Software** | | |
| LAS-AF (version 2009) | Leica | N/A |
| ImageJ/Fiji 2.9.0 | https://imagej.net/software/fiji/ Schindelin et al, 2012, https://doi.org/10.1038/nmeth.2019. | N/A |
| **Other** | | |
| N/A | | |

## Methods and protocols

### Cell lines and plasmid transfections

HeLaK and Hek293T were cultured in DMEM supplemented with 10% FCS at 37 °C and 5% $CO_2$. pCS2-based expression plasmids were transfected into HeLaK cells with Lipofectamine 2000 according to the manufacturer's protocol and into Hek293T cells using a calcium phosphate-based method.

### RNA interference

*LUCIFERASE siRNA* served as negative control *(siCONTROL)*. HeLaK and Hek293T cells were transfected with with 50–100 nmol siRNA duplex (Eurofins Genomics) using RNAiMax or a calcium phosphate-based method, respectively. To increase target knock-down efficiency, siRNAs 1 and 2 were transfected simultaneously. Since depletion of Sgo1 or sororin causes premature loss of cohesion, mitotic arrest, and cell death, cells were transfected approximately halfway through of an 18–20 h thymidine block, then released and analyzed in the following mitosis. In all other cases, siRNA transfer was performed before synchronization procedures were applied.

### Cell cycle synchronizations

For synchronization in early S-phase, cells were treated with thymidine (2 mM) for 20 h. For experiments with ectopic expression of Pds5 variants, plasmid transfection was done 8–12 h prior to thymidine addition. Synchronization of cells in prometaphase was achieved by 90-min-incubation with taxol (0.2 µg/ml) 9 h after release from a thymidine block. Alternatively, i.e., without pre-synchronization with thymidine, taxol was added for 10 h to asynchronous cells 36 h after plasmid transfection. Transfected, prometaphase-arrested cells were analyzed by chromosome spreading and served as a source of chromatin-free Pds5a/b for Wapl/sororin-binding assays. For an arrest in the late G2-phase, cells released for 6 h from a thymidine block were supplemented with RO-3306 (8–10 µM) for a maximum of 6 h. To visualize early mitotic events, RO-3306 was washed out with pre-warmed media, and cells were fixed 20–30 min thereafter. Alternatively, G2 cells were released into taxol for 2 h to re-arrest them in prometaphase.

### Antibodies

Non-commercial antibodies used in this study were generated at Charles River Laboratories. To this end, antigenic peptides (Bachem) were coupled via terminal Cys to Maleimide-activated KLH prior to immunization. Antibodies were affinity-purified against immobilized antigens. Antigenic proteins were coupled to NHS-activated sepharose, whereas antigenic peptides were coupled to SulfoLink coupling gel. For immunoprecipitation experiments, the following affinity matrices and antibodies were used: Mouse anti-Flag M2-Agarose, anti-GFP nanobody covalently coupled to NHS-agarose and guinea pig anti-Rad21 and guinea pig anti-Pds5b coupled to protein A sepharose. For non-covalent coupling of antibodies to beads, 10 µl of protein A sepharose were rotated with 2–5 µg antibody for 90 min at room temperature and then washed three times with 1xPBS, 1% BSA (w/v; Roth).

### Immunoprecipitation (IP)

About $2 \times 10^7$ cells were lysed with a dounce homogenizer in 2 ml lysis buffer 2 [20 mM Tris-HCl, pH 7.7; 100 mM NaCl; 10 mM

NaF; 20 mM β-glycerophosphate; 10 mM MgCl₂; 0.1% Triton X-100; 5% glycerol; 1x complete protease inhibitor cocktail], and incubated on ice for 10 min. To preserve phosphorylations, lysis buffer 2 was additionally supplemented with calyculin A (50 nM). Corresponding whole cell lysates were centrifuged for 20 min at 4 °C and 16,000×g to fractionate them into cleared lysates (soluble supernatant) and pelleted chromatin. For IP of soluble proteins, 10 μl of antibody-loaded beads were rotated with 2 ml cleared lysate for 4–12 h at 4 °C, washed 5x with lysis buffer 2, and equilibrated in corresponding reaction buffers. For IP of DNA-associated proteins, pelleted chromatin was washed twice in lysis buffer 2 and digested for 1 h at 4 °C with benzonase (30 U/l) in 2 ml lysis buffer 2. Then, insoluble material was removed by centrifugation (5 min at 4 °C and 2500×g), and the supernatant was combined with 10 μl of antibody-loaded beads and rotated for 4–12 h at 4 °C. Thereafter, beads were washed 6x with lysis buffer 2 supplemented with additional 250 mM NaCl before they were used for further applications.

### Immunofluorescence microscopy of spread chromosomes

HelaK transfected with the indicated siRNAs were synchronized in G2-phase by the addition of RO-3306 followed by subsequent release into taxol. Prometaphase-arrested cells were collected by shake-off 30 min thereafter and immediately processed. Following their pelleting (300×g, for 3 min), cells were resuspended in hypotonic buffer I (50 mM sucrose, 30 mM TRIS-HCl pH 8.2, 17 mM sodium citrate trihydrate, 0.2 μg/ml taxol), incubated for 7 min, sedimented again and resuspended in hypotonic buffer II (100 mM sucrose). Immediately thereafter, 5 μl cell suspension were dropped onto a coverslip wetted by immersion in fixation buffer (1% PFA, 5 mM sodium borate pH 9.2, 0.15% TritonX-100) and spread by tilting of the coverslip. After drying at RT, coverslips were washed 3x with PBS, blocked for 1 h at RT in PBS; 3% (w/v) BSA, and incubated with primary antibodies for 1 h in a wet chamber. Coverslips were then washed 4x with PBS-Tx (PBS; 0.1% Triton X-100), incubated with fluorescently labeled secondary antibodies for 40 min, washed again 4x and finally mounted in 20 mM Tris-HCl, pH 8.0; 2.33% (w/v) 1,4-diazabicyclo(2.2.2) octane; 78% glycerol on a glass slide. Samples were analyzed on a DMI 6000 inverted microscope (Leica) using a HCX PL APO 100x/1.40-0.70 oil objective collecting Z-stacks series over 2 μm in 0.2 μm increments. Images were deconvoluted and projected into one plane using the LAS-AF software. Smc2 staining was used to identify mitotic chromosomes. Standard chromosome spreads were done as described (McGuinness et al, 2005) and observed on an Axioplan 2 microscope (Zeiss) using a Plan-APOCHROMAT 100x/1.40 oil objective. A cell was counted as suffering from PCS when >50% of its chromosomes were separated into single chromatids, and as defective in prophase pathway signaling when >50% of its chromosomes exhibited zipped arms. At least 200 spreads were counted per condition.

### Isolation of crude chromatin

Crude G2 chromatin was freshly prepared for each experiment according to a protocol by Mendez and Stillman (Mendez and Stillman, 2000). Briefly, 4 × 10⁷ G2-arrested HelaK cells were resuspended in 3 ml buffer A [10 mM HEPES, pH 7.9; 10 mM KCl; 1.5 mM MgCl₂; 0.34 M sucrose; 10% glycerol; 1 mM DTT; 1x EDTA-free complete protease inhibitor cocktail] and lysis was initiated by addition of TritonX-100 to 0.1%. Following a 5 min incubation on ice, chromatin was pelleted by centrifugation for 4 min at 4 °C and 1300×g, washed once in TritonX-100-supplemented (0.1%) buffer A, re-isolated and resuspended in 3 ml buffer B (3 mM EDTA; 0.2 mM EGTA; 1 mM DTT; 1 mM spermidine; 0.3 mM spermine; 1x complete protease inhibitor cocktail). After incubation for 25 min on ice, the chromatin was sedimented for 4 min at 4 °C and 1700×g, washed once in 1 ml buffer B, and finally resuspended in 200 μl chromatin buffer (5 mM Pipes, pH 7.2; 5 mM NaCl; 5 mM MgCl₂; 1 mM EGTA; 10% glycerol).

### DNA-mediated chromatin pull-down

The following procedure is a combination of published protocols (Aranda et al, 2019, Kliszczak et al, 2011, Neef and Luedtke, 2011). Approximately 20 × 10⁷ HeLaK cells pre-synchronized by thymidine treatment (2 mM for 18 h) were released to undergo S-phase in the presence of F-ara-Edu (5 μM) for 5 h. Then, RO-3306 (10 μM) was added, and the G2-arrested cells were harvested and PBS-washed 5 h later. The cell pellet was immediately resuspended in 20 volumes of lysis buffer 1 [20 mM HEPES, pH 7.5; 0.22 M sucrose; 2 mM MgCl₂; 0.5% (v/v) Nonidet P-40; 0.5 mM DTT; 1x EDTA-free complete protease inhibitor cocktail], dounce homogenized and incubated for 10 min on ice. The whole cell lysate was then centrifuged for 15 min at 4 °C and 3000×g in a swing-out rotor to pellet G2 chromatin/nuclei. After removal of the supernatant, the chromatin pellet was supplemented (in this order) with (+)-sodium-L-ascorbate (10 mM), azide-PEG3-biotin (0.1 mM), and cooper(II)sulfate (2 mM) and slowly rotated in the dark for 30 min at room temperature (RT). This was followed by three consecutive wash steps (15 min at 4 °C and 3000×g in a swing-out rotor) with 1 ml lysis buffer 1 each to remove unbound biotin. To increase purity, resuspended chromatin was centrifuged through a 3 cm cushion (20 mM HEPES, pH 7.5; 2 mM MgCl₂; 0.34 M sucrose) for 20 min at 4 °C and 3400×g in a swing-out rotor. The resulting chromatin pellet was finally resuspended in 300 μl lysis buffer 1. High-capacity streptavidin beads (250 μl) were pre-blocked with 0.4 mg/ml sonicated salmon sperm DNA in lysis buffer 1 for 30 min at RT, washed twice (centrifugation: 1 min at 4 °C and 120×g) and then combined with the biotinylated G2 chromatin in a final volume of 4 ml. Where indicated, benzonase (30 U/l) was added. After 40 min of rotating at 4 °C, beads were washed once with lysis buffer 1 and then resuspended in cold high salt buffer (HSB; 20 mM HEPES, pH 7.5; 0.6 M NaCl; 2 mM MgCl₂; 0.22 M sucrose; 0.2% NP-40; 1 mM DTT). Following a 10-min incubation, the salt-extracted chromatin was very carefully washed twice with HSB and twice with lysis buffer 1 (using cut-off tips). Finally, the chromatin-loaded streptavidin beads were resuspended in 1 ml nuclear storage buffer [20 mM TRIS-HCl, pH 8.0; 75 mM NaCl; 0.5 mM EDTA; 50% (v/v) glycerol; 1 mM DTT; 1x EDTA-free complete protease cocktail] and stored at −20 °C.

### Recombinant protein expression and purification

Active Cdk1-cyclin B1 was expressed in insect cells and purified as described (Gorr et al, 2005). Active Plk1 was purchased (see resource table). To generate active aurora B kinase, the complex of co-expressed fragments of human aurora B kinase (amino acids 65–344) and INCENP (amino acids 819–918) was affinity-purified from bacteria essentially as described (Sessa, Mapelli et al, 2005).

Active Nek2a (active or kinase-dead, KD = Lys37Met) and Cdk1/2-cyclin A2 were expressed and purified as described (Hellmuth and Stemmann, 2020). Prior to use, all kinases were tested for activity using 2 µg each of histone H1 or Myelin Basic Protein (MBP) as model substrates. Human Wapl was purified from $5 \times 10^7$ transiently transfected Hek293T cells. To enrich the pool of soluble Wapl, cells were treated with taxol for 36 h after plasmid transfection and for 12 h to arrest them in prometaphase prior to harvesting. Following lysis of the cells in 10 ml lysis buffer 2 and clearing of the lysate by centrifugation (30 min at 4 °C and 18,000×$g$), GFP-SUMOstar-Wapl was immunoprecipitated for 90 min with 50 µl anti-GFP nanobody beads. Then, the beads were washed 5x with lysis buffer 2, transferred into cleavage buffer [10 mM Hepes-KOH, pH 7.7; 50 mM NaCl; 25 mM NaF; 1 mM EGTA; 20% glycerol] and finally rotated for 30 min at 18 °C with 5 µg His$_6$-SUMOstar protease. The corresponding eluate (50 µl) was snap-frozen in aliquots and stored at −80 °C. His$_6$-SUMO3-sororin was expressed from a pET28-derivative in *E. coli*. Rosetta 2 DE3. Bacteria were lysed in phosphate-buffered high saline (PBHS; 10 mM Na$_2$HPO$_4$; 2 mM KH$_2$PO$_4$; 500 mM NaCl; 2.7 mM KCl; 5 mM imidazole; 0.5 mM DTT), purified over Ni$^{2+}$-NTA-Agarose according to standard procedure and eluted in PBHS supplemented with imidazole to 250 mM and pH-adjusted with HCl to 7.5. Following its dialysis into sororin storage buffer (20 mM Tris-HCl, pH 6.9; 100 mM NaCl), His$_6$-SUMO3-sororin was stored in aliquots at −80 °C.

### In vitro cohesin release assay

To assemble kinase reactions, 15 µl chromatin (crude or immobilized on streptavidin sepharose) equilibrated in kinase buffer [10 mM Hepes-KOH, pH 7.7; 50 mM NaCl; 25 mM NaF; 1 mM EGTA; 20% glycerol; 10 mM MgCl$_2$; 5 mM MnCl$_2$; 5 mM DTT] were combined with ATP (ad 1 mM), 0.4 µg Cdk1-cyclin B1, 0.1 µg aurora B-INCENP, 0.1 µg Plk1, 1 µl Nek2a, and/or 1 µl Cdk1/2-cyclin A2, as indicated. The sample shown in lane 9 of Fig. 2C was additionally supplemented with okadaic acid (1 µM). The samples, which had a total volume of 25–30 µl, were incubated for 20–30 min at 37 °C while slowly rotating. This was followed by the addition of RO-3306 (2 µM), BI2536 (100 nM), staurosporine (300 nM), and/or ZM-447439 (0.5 µM), as indicated. After 10 min at RT, reactions were supplemented with 2 µl Wapl and incubated for 20 min at 37 °C while slowly rotating. Then, the chromatin was pelleted for 1 min at 4 °C and 1700×$g$, and the supernatant was recovered for immunoblotting. The chromatin was washed twice with 1 ml each of lysis buffer 2, re-pelleted for 4 min at 1700×$g$, and finally resuspended in 20 µl 2x SDS-sample buffer for analysis. In the release assay performed in Fig. 8B, the supernatant of the kinase reaction was recovered ("supernatant 1"). Chromatin beads were then washed twice with 1 ml each of cleavage buffer and resuspended in 30 µl release buffer [30 mM Hepes-KOH, pH 7.7; 200 mM NaCl; 25 mM KCl; 1 mM EGTA; 30% glycerol; 5 mM MgCl$_2$; 1 mM DTT] and supplemented with ATP (ad 1 mM) and/or 2 µl Wapl. The rest of the procedure was the same as described above.

### Kinase and binding assays

For phosphorylation of immobilized Flag-Pds5 and Rad21-IPs, 15 µl beads (instead of chromatin) were used to assemble kinase reactions essentially as described above. For the samples shown in lanes 3, 4, 6, and 7 of Fig. 6A, samples were additionally supplemented with calyculin A (20 nM). Samples were incubated for 20–30 min at 37 °C while slowly rotating. If the kinase reactions were directly analyzed by immunoblotting, then the beads were pelleted for 1 min at 4 °C and 1700×$g$, washed once with 1 ml lysis buffer 2, re-pelleted, and eluted by boiling in 15 µl 2x SDS-sample buffer. To assess sororin/Wapl binding to Pds5, Flag-IPs (directly from cells or after in vitro-phosphorylation) were washed twice in cleavage buffer containing kinase inhibitors, resuspended in 15 µl cleavage buffer containing 2 µl Wapl and 0.3 µg sororin and rotated for 20 min at 37 °C. Then, the supernatant was recovered for later immunoblotting and the beads were washed twice with 1 ml each of lysis buffer 2 before they, too, were eluted by boiling in SDS-sample buffer. For radioactive kinase assays, Rad21, Smc1α, Smc3, Stag1, Stag2, and Pds5b variants were first expressed by "cold" coupled in vitro transcription-translation (IVTT). IVTT aliquots (3 µl) were then incubated for 20–30 min at 37 °C with γ-$^{33}$P-ATP (40 µCi, Hartmann Analytic), "cold" ATP (5 µM) and the indicated kinases in a total volume of 25–30 µl (see above). To specifically inhibit kinase activity during the incubation at 37 °C, the corresponding inhibitor (see above) was added together with the kinase during reaction assembly at 4 °C. Following SDS-PAGE, radioactive reactions were blotted onto PVDF membrane and first analyzed by autoradiography. The same membrane was then re-activated with methanol and further examined by immunoblotting.

## Data availability

The authors declare that all data supporting the findings of this study are available within the paper and its supplementary information files. If there is reasonable request data can also be provided from the corresponding author.

The source data of this paper are collected in the following database record: biostudies:S-SCDT-10_1038-S44318-024-00228-9.

## Peer review information

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

## Acknowledgements

We thank B. Neumann for recombinant sororin and aurora B kinase and S. Heidmann, T. Klecker, and J. Michl for critical reading of the manuscript. This work was supported by a grant (STE997/8-1) from the Deutsche Forschungsgemeinschaft (DFG) to OS.

## Author contributions

**Susanne Hellmuth**: Conceptualization; Investigation; Methodology; Writing— original draft; Writing—review and editing. **Olaf Stemmann**: Conceptualization;

Data curation; Supervision; Funding acquisition; Writing—original draft; Writing—review and editing.

Source data underlying figure panels in this paper may have individual authorship assigned. Where available, figure panel/source data authorship is listed in the following database record: biostudies:S-SCDT-10_1038-S44318-024-00228-9.

## Funding

## Disclosure and competing interests statement

The authors declare no competing interests.

# Expanded View Figures

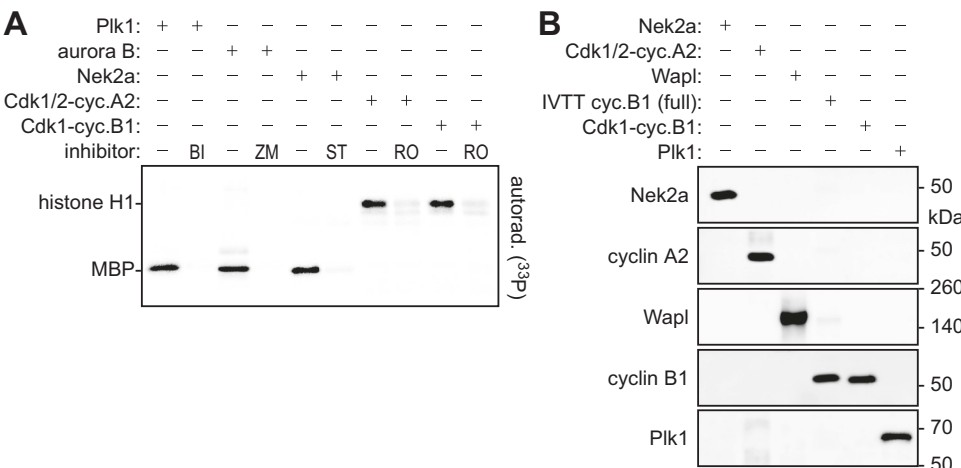

**Figure EV1.  Characterization of recombinant kinases.**

(**A**) Recombinant kinases are active as judged by the phosphorylation of model substrates. Plk1, aurora B-INCENP, Nek2a, Cdk1/2-cyclin A2, or Cdk1-cyclin B1 supplemented with their specific inhibitor or carrier solvent DMSO (−) were incubated with the corresponding model substrate in the presence of [γ$^{33}$P]-ATP. Reactions were subjected to SDS-PAGE followed by autoradiography. BI BI2536, ZM ZM-447439, ST staurosporine, RO RO-3306, MBP myelin basic protein. (**B**) Recombinant Nek2a, Cdk1/2-cyclin A2, and Wapl are free of cyclin B1 and Plk1. The preparations of Nek2a, Cdk1/2-cyclin A2, Wapl, Cdk1-cyclin B1, and Plk1 used for the release assays were characterized by immunoblotting using the indicated antibodies. In vitro expressed (IVTT) cyclin B1 served as an additional control.

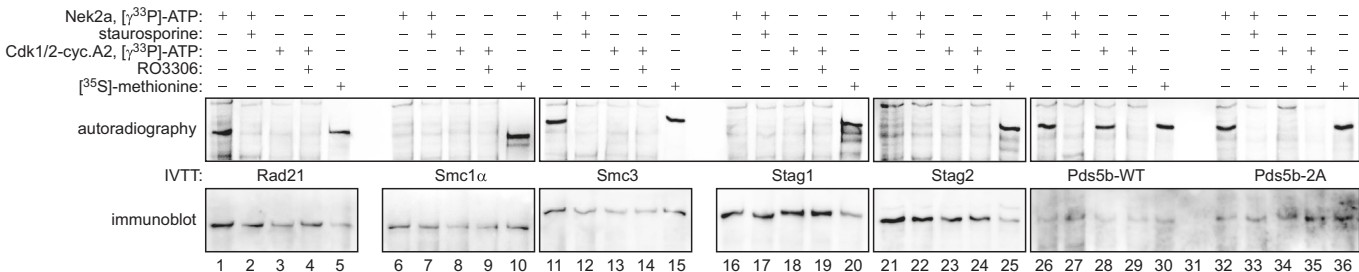

**Figure EV2. Nek2a and Cdk1/2-cyclin A2 both phosphorylate in vitro expressed Pds5b.**

Cohesin subunits and Pds5b variants were expressed by coupled in vitro transcription-translation (IVTT), incubated with Nek2a, Cdk1/2-cyclin A2, [γ-³³P]-ATP, and kinase inhibitors, as indicated, separated by SDS-PAGE and analyzed by autoradiography and immunoblotting. ³⁵S-methionine labeled IVTT products served as indicators of the migration behavior of the respective full-length protein. Note that Pds5b-2A (Ser1161,1166Ala) is resistant to phosphorylation by Cdk1/2-cyclin A2 but not Nek2a.

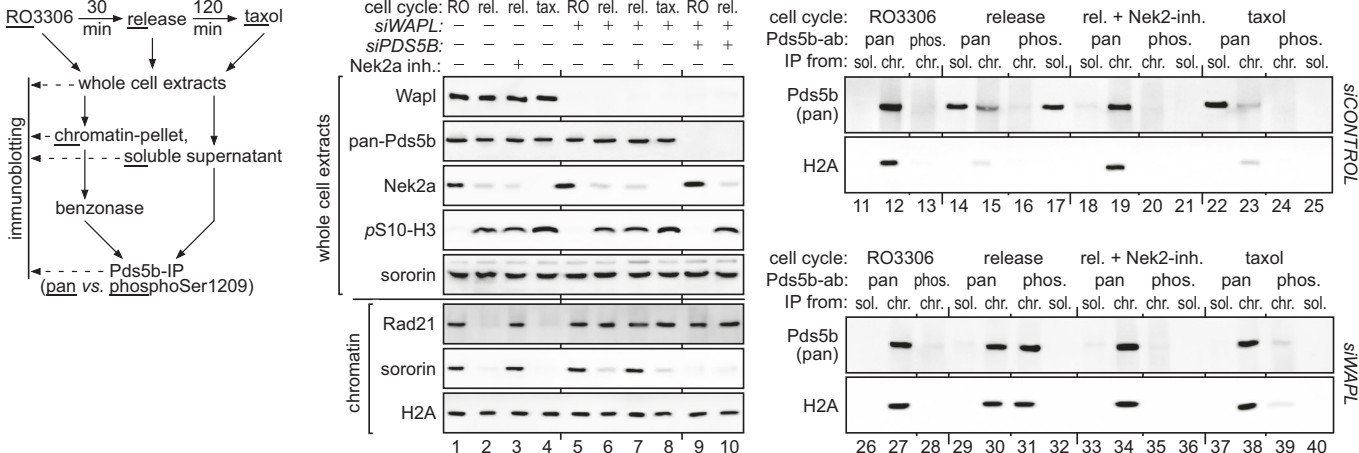

**Figure EV3.  Pds5b phosphorylated on Ser1209 by Nek2a exhibits Wapl-dependent displacement from early mitotic chromatin.**

HeLaK cells transfected to deplete Pds5b and/or Wapl by RNAi were synchronized and harvested in G2-, pro-, and prometaphase, fractionated into chromatin and soluble lysate and subjected to the indicated (IP-) Western analyses. Nek2a inhibitor (NCL 00017509) was added at the time of release from RO-3306 and cells were harvested 30 min thereafter.

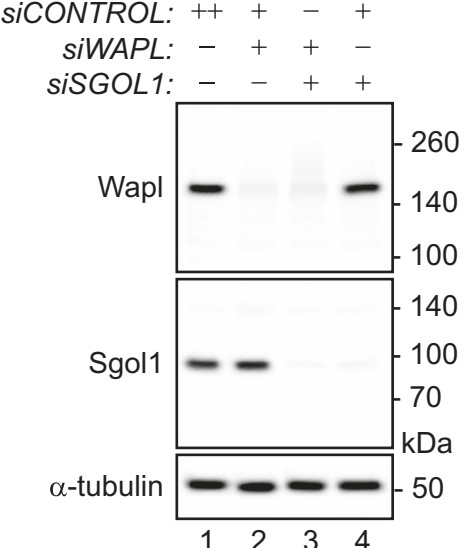

**Figure EV4.  Wapl and/or Sgo1 are efficiently depleted by RNAi.**

HeLaK cells transfected with the indicated siRNAs during a thymidine arrest were released into a G2- and, from there into a prometaphase arrest prior to their analysis by immunoblotting. Aliquots of the cells analyzed in lanes 2 and 3 were subjected to spread-IFs as shown in Fig. 6B (lower row panels).

