## [Peer Review File · The EMBO Journal]

Requirement of Nek2a and cyclin A2 for Wapl-dependent removal of cohesin from prophase chromatin

Susanne Hellmuth and Olaf Stemmann

Corresponding author(s): Susanne Hellmuth (susanne.hellmuth@uni-bayreuth.de)

Review Timeline:

Submission Date:	8th Mar 24
Editorial Decision:	12th Apr 24
Revision Received:	27th Jun 24
Editorial Decision:	9th Aug 24
Revision Received:	14th Aug 24
Accepted:	27th Aug 24

Editor: Hartmut Vodermaier

Transaction Report:

Dr. Susanne Hellmuth
University of Bayreuth
Chair of Genetics
Universitaetsstrasse 30
Bayreuth 95440
Germany

12th Apr 2024

Re: EMBOJ-2024-117218
Requirement of Nek2a and cyclin A2 for Wapl-dependent removal of cohesin from prophase chromatin

Dear Dr. Hellmuth, dear Olaf,

Thank you for submitting your study on early mitotic APC/C targets regulating Wapl-dependent cohesin removal during prophase. I sent it to three expert referees, who have now returned the reports copied below. As you will see, all referees acknowledge the interest and potential importance of your findings, but there are also a number of substantive issues (especially in the comments of referee 2) that would require further clarification prior to publication. Should you be able to adequately address these concerns, as well as the various more specific points listed, we would be happy to consider a revised version of the study further for The EMBO Journal.

Since it is our policy to aim for only a single round of major revision, it will be important to diligently respond to each referee point at the time of resubmission, and I would therefore encourage you to contact me with a preliminary point-by-point response already during the early stages of your revision work, in order to clarify how key issues may be solved and for us to agree on a revision plan. We would also be open to extension of the default three-months revision period if needed; our 'scooping protection' (meaning that competing work appearing elsewhere in the meantime will not affect our considerations of your study) would of course remain valid also throughout such an extension.

Further information on preparing, formatting and uploading a revised manuscript can be found below and in our Guide to Authors. Thank you again for the opportunity to consider this work for The EMBO Journal, and I look forward to hearing from you in due time.

With kind regards,

Hartmut

9) Digital image enhancement is acceptable practice, as long as it accurately represents the original data and conforms to community standards. If a figure has been subjected to significant electronic manipulation, this must be clearly noted in the figure legend and/or the 'Materials and Methods' section. The editors reserve the right to request original versions of figures and the original images that were used to assemble the figure. Finally, we generally encourage uploading of numerical as well as gel/blot image source data; for details see: embopress.org/page/journal/14602075/authorguide#sourcedata

At EMBO Press, we ask authors to provide source data for the main manuscript figures. Our source data coordinator will contact you to discuss which figure panels we would need source data for and will also provide you with helpful tips on how to upload and organize the files.

In the interest of ensuring the conceptual advance provided by the work, we recommend submitting a revision within 3 months (11th Jul 2024). Please discuss the revision progress ahead of this time with the editor if you require more time to complete the revisions. Use the link below to submit your revision:

Link Not Available

Referee #1:

In this manuscript, Hellmuth and Stemmann ask how pericentromeric cohesin is protected from Wapl-mediated release in prometaphase, when Sgo1 relocates to the inner centromere after sensing the tension of kinetochore-bound microtubules. They use biochemical reconstitution of the Wapl-dependent cohesin removal reaction to uncover a role for Nek2 and CDK1-CyclinA2 in phosphorylation of Pds5B that is required for this removal. As these kinases are inactivated by prometaphase, the fraction of cohesive cohesin remaining at this time should not be released by Wapl, only by separase.

This is a very interesting study, with beautiful biochemical experiments, carefully designed and controlled, that overall support the conclusions of the authors. These experiments are validated with in vivo data looking at chromosome cohesion. The paper is a bit dense but the authors make every effort to guide readers by including in most Figures the experiment outline.

I have some questions/suggestions to strengthen the manuscript and satisfy my curiosity, but other than that, my only concern is Pds5A. Previous results in HeLa (Losada 2005 J Cell Sci) and MEFs (Carretero 2013 EMBO J) suggest a more important role of Pds5B in centromeric cohesion. The basis for this specificity is unclear. Here, the authors identify and mutate phosphorylation sites in Pds5B (for Nek2 and for CDK1-Cyclin A2) and demonstrate how these phosphorylations favor Pds5B binding to Wapl over binding to Sororin, thus promoting cohesin release in prophase. They also show that these sites are not conserved in Pds5A and the two kinases have no effect on the binding behavior of Pds5A towards Wapl and Sororin. It is therefore unclear what happens to the fraction of cohesin complexes that contain Pds5A instead of Pds5B. In fact, in order to show the effect of Pds5B phosphorylation mutants on sister chromatid cohesion (Figure 5) the authors first deplete endogenous Pds5A and Pds5B. What happens if only Pds5B is depleted? This would help us understand the contribution of the proposed pathway to regulation of

cohesion and the potential differences between Pds5A and Pds5B.

Additional questions/issues:

-In the *in vitro* reconstitution of the prophase pathway (Figure 1), I find strange that the high salt wash (0.6M NaCl) strips histones but not HP1a or even Pds5B. Any thoughts? Also, could you check what happens to Pds5A and Sgo1? What about endogenous Wapl, what is the excess of Wapl that must be added to promote the release in this *in vitro* assay?

-In the description of the assay in Methods:

"To assemble kinase reactions, 15 μ l chromatin (crude or immobilized on streptavidinsepharose) equilibrated in kinase buffer [...] were combined with ATP (1 mM), okadaic acid (1 μ M, Alexis Biochemicals) or calyculin A (20 nM, LC Laboratories) and with 0.4 μ g Cdk1-cyclin B1, 0.1 μ g aurora B-INCEP, 0.1 μ g Plk1, 1 μ l Nek2a and/or 1 μ l Cdk1/2-cyclin A2, as indicated". From this description it seems that either okadaic acid or calyculin A are included in the reaction buffer by default, as ATP. Is this correct?

-It would be nice to show that Sgo1 is responsible for keeping Pds5B dephosphorylated (Figure 4) by including double siWAPL siSgo1 condition.

-In experiment shown in Figure 5, a phosphomimetic mutant is used in which six Ser are mutated to Asp. Please explain why site 1162 is mutated.

-I encourage the authors to add a model/graphical summary of conclusions that includes kinases and residues modified.

Referee #2:

Sister chromatid cohesion is essential for equal chromosome segregation in eukaryotes. Upon entry into mitosis, most of cohesion in chromosome arm region is dissolved by a cohesin dissociation factor Wapl via so-called prophase pathway, while pericentromeric cohesion is protected by Sgo1-PP2A. Although mitotic phosphorylations on cohesin and Sororin by Plk1, AuroraB, and Cdk1-cyclin B are known to facilitate mitotic cohesin dissociation from chromosome arms, their spatial and temporal regulations are not yet well understood. Especially, how pericentromeric cohesion is protected after Sgo1-PP2A is relocated from pericentromere to inner kinetochore well before anaphase onset is yet enigmatic.

In this manuscript, by establishing *in vitro* reconstitution system of prophase pathway, Hellmuth and Stemmann found that two kinases Nek2a and Cdk1/2-cyclin A2 are required for Wapl-dependent cohesin unloading and identified these kinases phosphorylate Pds5b, a cohesin binding factor, on its C-terminus. The phosphorylation promoted Pds5b's dissociation from Sororin and association with Wapl. By generating the phospho-deficient and phospho-mimetic mutants, the authors showed that phosphor-deficient mutants stabilized cohesin and Sororin on chromatin, and oppositely non-degradable cyclin A2 and phosphor-mimetic mutants induced premature sister chromatid separation.

Their findings that Nek2a and Cdk1/2-cyclin A2 directly phosphorylate Pds5b and that the phosphorylations convert Pds5b from a Sororin to a Wapl binder are truly novel and important for chromosome study field. In addition, *in vitro* reconstitution of prophase pathway is remarkable. However, it is still too premature to conclude that Plk1 and Cdk1-cycB1 are not required for prophase pathway only based on the reconstitution system because it is less clear to what extent their reconstitution system recapitulate physiological condition and also the reconstitution assay is not performed in a quantitative manner. Because the Nek2a and Cdk1/2-cyclin A2 kinases are novel regulators in sister chromatid cohesion mechanism, more detailed descriptions of Nek2a- and Cdk1/2-cyclin A2-dependent phosphorylation would be needed.

Specific major concerns:

1) In the reconstitution experiments in the figure 1 and 2, protein purities and kinase activities are not well described. The authors should show Coomassie staining of purified proteins that they added (also in Fig EV4), and also describe their final concentrations. In addition, kinase activities are not quantitatively tested. Though it is shown that all Nek2a, Cdk1/2-cyclin A2, and Cdk1-cyc B1 are somewhat active (Figure EV1A), it should be shown in more quantitative way (e.g. showing the time-course of substrate phosphorylation).

2) To see to what extent the reconstitution system sensitively recapitulates cellular events, it would be testable how much chromatin bound proteins are dissociated or stabilized after siRNAs for Sororin, Escp1/2, or Wapl. For instance, do cohesin subunits (Smc3, Rad21, or Stag1) still stay on chromatin after Sororin RNAi? If yes, are these cohesin complexes still resistant to kinase treatment without Wapl?

3) In Figure EV3B, it is shown that pS1209 phosphorylation appears only in soluble fraction in released sample (lane 17). However, it is unclear when and where the phosphorylation by Nek2a and Cdk1/2-cycA2 occurs during cell cycle. The authors should describe this.

4) In Figure 2B, the authors should include negative controls, where wild-type Pds5b were bound to the beads with or without

Nek2a and Cdk1/2-cyclin A2. Though the result might be evident from previous results, if the Pds5b purification was done in the different way from different cells, then negative controls are always needed in each experiment.

5) In Figure 3B, what is the consequence if both Nek2a Δ MR and Δ 86-cyclin A2 were expressed with Pds5-5A mutant?

6) In Figure 5A, it is shown that Pds5b-5A binds to Sororin but not Wapl in prometaphase in whole cell extract. How about chromatin fraction? Is the association between Sororin and Pds5b-5A found on chromatin too?

Minor point:

7) In Figure 2B, headings of Pds5b's 2A, 3A and 5A are confusing. Please indicate 2A is from lane 6 to 8, 3A is from lane 9 to 11, and 5A is from lane 12 to 14.

Referee #3:

In this manuscript Helmuth and Stemmann have provided a very careful analysis of the mechanisms that result in cohesin removal from chromosome arms during mitotic prophase. Here they focus on Pds5B, and the competition between Wapl and Sororin for Pds5B binding, and cohesin release in prophase. The data are thorough and offer important insight into how this mechanism is regulated in vertebrate models. It should be published. I strongly recommend efforts to improve the readability.

Major concerns:

1. The writing is extremely dense and at times non-linear, and should be streamlined and possibly shortened, to make the manuscript more readable. For example, figure EV1 first shows kinase assays before telling us why they are being done. Better to introduce all kinases as possible modulators of the pathway, then walk through the data in EV1B (this would prevent going back and forth EV1A, EV1B, EV1A, EV1B, etc. making the writing more readable). More paragraph breaks to define each question/conclusion in the results section would also have been helpful. Some of the details in the results section should be moved to the figure legends, some of which are very minimal. Detailed EV panel lane callouts in the main text probably aren't necessary. Trust the reader! I don't know if the authors were panel-limited in the main figures, but it might help to move some of the EV data to the main figures.

2. A cartoon or table summarizing the various P-site mutants in the Pds5B tail, with the kinases thought to modify the residues (or color code the residues), and the mutant names would be great.

3. A cartoon explaining their model, integrating what is known about Pds5A vs Pds5B, centromeres vs arms, would be very useful.

4. Discussion: The discussion of Pds5A is important, and a bit confusing. The phosphorylation sites identified and characterized in Pds5B are not present in Pds5A, and many of their experiments are done in the absence of Pds5A. Their findings are consistent with previous observations in the literature (centromere vs arm cohesion), but it would have been helpful if they could discuss the relative roles of Pds5A and Pds5B and the impacts of their differences in regulation, if any, on mitotic chromosome structure and function. The discussion of Pds5A seems to be suggesting that Pds5A is similarly controlled by Nek2a, but not through phosphorylation of the same residues, or perhaps through modification of an additional protein.

Minor concerns (in no particular order):

1. Can they discuss inter-sister vs loop cohesion in their reconstitution system. Do they need to use G2 chromatin for the release assay to work? If so, why? Are they really just assaying inter-sister cohesion? Is this Pds5B-dependent mechanism uniquely involved in sister cohesion in prophase?

2. Introducing potential roles of Nek2 and cyclin A in the prophase pathway in the introduction would have been helpful.

1. Not sure what is meant by Cdk1/2 when you are talking about a specific reagent. Can you please explain? Was this cyclin IP with mixed kinases?

2. First line in the Results section, should read from (says "form")

3. The use of "see below" is confusing. Please indicate exactly where this is shown, or change the order of the callouts in the text.

4. "Resistant against" should be "resistant to" (results page 2).

5. "Surprisingly, in the presence of Wapl, Nek2a, and Cdk1/2-cyclin A2, aurora B-INCENP was necessary and sufficient for removal of cohesin and sororin from chromatin..." This is a strange construction. How about "was able to ..., while Cdk/cyclin B was not"? Sufficiency is not really the appropriate term when you have wapl, nek2a and cdk/cyclin in the reaction.

6. Please provide a reference or description of the "stabilized truncation variants" of Nek2a and cyclin

7. What are HeLaK cells?

8. Numerous spelling errors, including argueing, agrevated. Please check thoroughly.

9. How does this fit with the initial observation that Plk-dependent phosphorylation of SA2 causes cohesin removal (Hauf et al 2005)?

10. It would be nice to have a table for antibodies and reagents, instead of a long form paragraph.

Point-by-point response (in blue) to referees' comments (in black)

Referee #1:

In this manuscript, Hellmuth and Stemmann ask how pericentromeric cohesin is protected from Wapl-mediated release in prometaphase, when Sgo1 relocates to the inner centromere after sensing the tension of kinetochore-bound microtubules. They use biochemical reconstitution of the Wapl-dependent cohesin removal reaction to uncover a role for Nek2 and CDK1-CyclinA2 in phosphorylation of Pds5B that is required for this removal. As these kinases are inactivated by prometaphase, the fraction of cohesive cohesin remaining at this time should not be released by Wapl, only by separate.

This is a very interesting study, with beautiful biochemical experiments, carefully designed and controlled, that overall support the conclusions of the authors. These experiments are validated with in vivo data looking at chromosome cohesion. The paper is a bit dense but the authors make every effort to guide readers by including in most Figures the experiment outline.

We thank referee #1 for this positive overall assessment of our work.

I have some questions/suggestions to strengthen the manuscript and satisfy my curiosity, but other than that, my only concern is Pds5A. Previous results in HeLa (Losada 2005 J Cell Sci) and MEFs (Carretero 2013 EMBO J) suggest a more important role of Pds5B in centromeric cohesion. The basis for this specificity is unclear. Here, the authors identify and mutate phosphorylation sites in Pds5B (for Nek2 and for CDK1-Cyclin A2) and demonstrate how these phosphorylations favor Pds5B binding to Wapl over binding to Sororin, thus promoting cohesin release in prophase. They also show that these sites are not conserved in Pds5A and the two kinases have no effect on the binding behavior of Pds5A towards Wapl and Sororin. It is therefore unclear what happens to the fraction of cohesin complexes that contain Pds5A instead of Pds5B. In fact, in order to show the effect of Pds5B phosphorylation mutants on sister chromatid cohesion (Figure 5) the authors first deplete endogenous Pds5A and Pds5B. What happens if only Pds5B is depleted? This would help us understand the contribution of the proposed pathway to regulation of cohesion and the potential differences between Pds5A and Pds5B.

We appreciate the suggestion to explore the effects of our Pds5b phosphorylation site variants in the background of single depletions of endogenous Pds5a or Pds5b. However, Losada and coworkers clearly showed that arm cohesion is mediated by both Pds5a and Pds5b, while (peri)centromeric cohesion is mediated only by Pds5b. A corollary is that Pds5a-cohesin is clearly responsive to Wapl-dependent removal in prophase of mitosis. (Otherwise, one would not observe separated arms in metaphase chromosome spreads.) Thus, it is highly likely that we would observe the same phenotypes seen after expression of the Pds5b mutants in the individual depletions as in the double knock-down - with one exception: In the mere absence of Pds5a, endogenous Pds5b should rescue the premature chromatid separation (PCS) phenotype caused by Pds5B-6D. Because the effort would be very high but we would likely learn very little, we refrained from doing this experiment.

However, we have prepared fresh 'click-DNA' (see next point) and conducted additional release assays. These experiments clearly show that, much like Pds5b, Pds5a is associated with G2 chromatin and evicted upon treatment with Nek2a, Cdk1/2-cyclin A2, aurora B-INCEP, Wapl and ATP (new Figure 8B). Thus, while Pds5a on its own is not directly affected by Nek2a and Cdk1/2-cyclin A2 in its binding behavior towards sororin and Wapl, Pds5a-cohesin is nevertheless susceptible to Wapl-dependent release. We cannot yet explain these observations but speculate in the discussion as follows (page 11): "Since Aurora B-dependent phosphorylations appear to be insufficient for cohesin release (Figure 1B), Nek2a and/or Cdk1/2-cyclin A2 are likely required also for opening of Pds5a-cohesin. This suggests the existence of an additional target of these kinases, with sororin being an

obvious candidate. Alternatively, Rad21 and/or Smc3, both of which are phosphorylated by Nek2a *in vitro* (Figure EV2), may be relevant substrates. This is because their NHD and neck region, respectively, represent cohesin's exit gate which is predicted to also make contact with sororin (Nasmyth et al., 2023)."

Additional questions/issues:

-In the *in vitro* reconstitution of the prophase pathway (Figure 1), I find strange that the high salt wash (0.6M NaCl) strips histones but not HP1 α or even Pds5B. Any thoughts? Also, could you check what happens to Pds5A and Sgo1?

In our experience, direct protein-DNA interactions are often salt-sensitive (due to the predominance of ionic bonds), whereas protein-protein interactions (which also form the basis for the topological linkage of cohesin to DNA) are often salt-resistant (due to higher contributions of hydrogen bonds, van der Waals bonds, and hydrophobic bonds). Therefore, the only real surprise to us was that the HP1 α -DNA interaction withstands higher salt than histones. As we prepared new G2 chromatin during the revision, we washed with increasingly higher salt, following the distribution of respective proteins between beads and supernatant at each step. Confirming previous results, we found that at 0.6 M NaCl histone H2A is quantitatively lost, while about half of cohesin, sororin, HP1 α and shugoshin is retained on the DNA beads (see figure below). We then took the beads from lane 4 (arrow head) as starting material for the new release assay (Figure 8B). If wanted, we could include the figure in the ms, but given its redundancy with figure 2B and the density of the data as it is, we would prefer to leave it out.

What about endogenous Wapl, what is the excess of Wapl that must be added to promote the release in this *in vitro* assay?

Unfortunately, we cannot currently answer this question of stoichiometry. Both the amount and purity of the G2 chromatin and Wapl are not sufficient to quantitatively compare them by Coomassie-staining.

-In the description of the assay in Methods:

" To assemble kinase reactions, 15 μ l chromatin (crude or immobilized on streptavidinsepharose) equilibrated in kinase buffer [...] were combined with ATP (1 mM), okadaic acid (1 μ M, Alexis Biochemicals) or calyculin A (20 nM, LC Laboratories) and with 0.4 μ g Cdk1-cyclin B1, 0.1 μ g aurora B-INCENP, 0.1 μ g Plk1, 1 μ l Nek2a and/or 1 μ l Cdk1/2-

cyclin A2, as indicated".

From this description it seems that either okadaic acid or calyculin A are included in the reaction buffer by default, as ATP. Is this correct?

Actually, no. Okadaic acid was included only in the experiment shown in figure 2C (of the revised ms) and calyculin A was used only in the experiment shown in figure 6A (of the revised ms). We apologize for not having made this clearer. In the revised manuscript, this has now been specified in the methods section as follows:

***In vitro* cohesin release assay**

To assemble kinase reactions, 15 μ l chromatin (crude or immobilized on streptavidin sepharose) equilibrated in kinase buffer [10 mM Hepes-KOH, pH 7.7; 50 mM NaCl; 25 mM NaF; 1 mM EGTA; 20% glycerol; 10 mM MgCl₂; 5mM MnCl₂; 5 mM DTT] were combined with ATP (ad 1 mM), 0.4 μ g Cdk1-cyclin B1, 0.1 μ g aurora B-INCENP, 0.1 μ g Plk1, 1 μ l Nek2a and/or 1 μ l Cdk1/2-cyclin A2, as indicated. The sample shown in lane 9 of figure 2C was additionally supplemented with okadaic acid (1 μ M, Alexis Biochemicals). The samples, which had a total volume of 25-30 μ l, were incubated for 20-30 min at 37°C while slowly rotating. This was followed by addition of RO-3306 (2 μ M, Santa-Cruz), BI-2536 (100 nM, Boehringer-Ingelheim), staurosporine (300 nM, Abcam) and/or ZM-447439 (0.5 μ M, Tocris), as indicated. After 10 min at RT, reactions were supplemented with 2 μ l Wapl and incubated for 20 min at 37°C while slowly rotating. Then,...

Kinase and binding assays

For phosphorylation of immobilized Flag-Pds5 and Rad21-IPs, 15 μ l beads (instead of chromatin) were used to assemble kinase reactions essentially as described above. For the samples shown in lanes 3, 4, 6 and 7 of figure 6A, samples were additionally supplemented with calyculin A (20 nM, LC Laboratories). Samples were incubated for 20-30 min at 37°C while slowly rotating....

-It would be nice to show that Sgo1 is responsible for keeping Pds5B dephosphorylated (Figure 4) by including double siWAPL siSgo1 condition.

We thank reviewer #1 for this excellent suggestion. Following her/his advice, we have repeated the spread-IF experiment from cells transfected with either siWAPL + siSGO1 or siWAPL alone and compared the (peri)centromeric pS1209-Pds5b signals. Indeed, the depletion of Sgo1 resulted in an uniform pS1209-Pds5b staining which now included pericentromeres. This experiment gave rise to the new figure EV4 and the lower panels in figure 6B.

-In experiment shown in Figure 5, a phosphomimetic mutant is used in which six Ser are mutated to Asp. Please explain why site 1162 is mutated.

Two considerations led us to do this: i) We had heard that some researcher use two directly juxtaposed Asp residues to mimic phosphorylation. ii) At this position of Pds5b one finds 4 Ser residues in a row (1159-1162); we originally thought that maybe two of those could be phosphorylated. However, in retrospect we suspect that this measure was unnecessary; the effect of the 5xAla variant suggests that a 5xAsp would probably have been sufficient....

-I encourage the authors to add a model/graphical summary of conclusions that includes kinases and residues modified.

We have improved the corresponding cartoon to now also include the kinase information (Figure 3B of the revised ms).

Furthermore, we have added the below model as part of the revised ms (new figure 1A):

Legend: Pericentromeric cohesin is deprotected long before sister chromatids separate. During phosphorylation- and Wapl-dependent release of cohesin from chromosome arms, pericentromeric cohesin is protected by associated Sgo1-PP2A. Following the early mitotic degradation of Nek2a and cyclin A2, Sgo1-PP2A relocates to kinetochores. Yet, pericentromeric cohesion persists until securin and cyclin B1 are degraded and separase is activated. A requirement for Nek2a and cyclin A2 for prophase pathway signaling would explain why in metaphase Wapl can no longer release cohesin.

Finally, we will show the below graphical summary as new figure 8C.

Legend: Model. In mitotic prophase Nek2a and Cdk1/2-cyclin A2 phosphorylate Pds5b within its unstructured C-terminal domain, thereby converting it from a sororin- to a Wapl-binder (i and ii). Sgo1-PP2A-dependent dephosphorylations preserve the Pds5b-sororin interaction at pericentromeres. Work from others suggest that ATP-dependent engagement of SMC heads then result in transient detachment of Rad21's NHD from Smc3's neck (Nasmyth et al., 2023). The same authors further proposed that sequestration of Rad21's N-terminus by Wapl stabilizes the open state of the gate, thereby giving DNA time to exit the cohesin ring (ii and iii). Note that DNA, SA1/2 and additional phosphorylations were omitted for sake of clarity.

Referee #2:

Sister chromatid cohesion is essential for equal chromosome segregation in eukaryotes. Upon entry into mitosis, most of cohesion in chromosome arm region is dissolved by a cohesin dissociation factor Wapl via so-called prophase pathway, while pericentromeric cohesion is protected by Sgo1-PP2A. Although mitotic phosphorylations on cohesin and Sororin by Plk1, AuroraB, and Cdk1-cyclin B are known to facilitate mitotic cohesin dissociation from chromosome arms, their spatial and temporal regulations are not yet well understood. Especially, how pericentromeric cohesion is protected after Sgo1-PP2A is relocated from pericentromere to inner kinetochore well before anaphase onset is yet

enigmatic.

In this manuscript, by establishing *in vitro* reconstitution system of prophase pathway, Hellmuth and Stemmann found that two kinases Nek2a and Cdk1/2-cyclin A2 are required for Wapl-dependent cohesin unloading and identified these kinases phosphorylate Pds5b, a cohesin binding factor, on its C-terminus. The phosphorylation promoted Pds5b's dissociation from Sororin and association with Wapl. By generating the phospho-deficient and phospho-mimetic mutants, the authors showed that phospho-deficient mutants stabilized cohesin and Sororin on chromatin, and oppositely non-degradable cyclin A2 and phospho-mimetic mutants induced premature sister chromatid separation. Their findings that Nek2a and Cdk1/2-cyclin A2 directly phosphorylate Pds5b and that the phosphorylations convert Pds5b from a Sororin to a Wapl binder are truly novel and important for chromosome study field. In addition, *in vitro* reconstitution of prophase pathway is remarkable.

We thank referee #1 for the positive assessment of our work up to this point.

However, it is still too premature to conclude that Plk1 and Cdk1-cycB1 are not required for prophase pathway only based on the reconstitution system because it is less clear to what extent their reconstitution system recapitulate physiological condition...

We totally agree that we cannot conclude that Plk1 and Cdk1-cyclin B1 are not required for Wapl-dependent release of cohesin *in vivo*. We did not intend to make this implication and apologize for not having made this clearer. In the revised manuscript we now try to correct this shortcoming by writing:

Page 4 (results): "... While this confirms the dispensability of Cdk1-cyclin B1 and Plk1 for cohesin release under the conditions of our *in vitro* assay, it does not necessarily contradict reports of their *in vivo* requirement for prophase pathway signaling (see discussion)."

Page 10 (discussion): "The dispensability of Plk1 and Cdk1-cyclin B1 for the *in vitro* release of cohesin from chromatin is surprising because several independent studies clearly showed the requirement of these two kinases for prophase pathway signaling in cells (Dreier et al., 2011, Gimenez-Abian et al., 2004, Lenart et al., 2007, Liu et al., 2013a, Losada et al., 2002, Nishiyama et al., 2013, Sumara et al., 2002). We cannot exclude that sites important for cohesin release are phosphorylated by recombinant Nek2a, Cdk1/2-cyclin A2 and/or aurora B *in vitro* but actually targeted by Plk1 and/or Cdk1-cyclin B1 *in vivo*. Alternatively, the dispensability of Plk1 and Cdk1-cyclin B1 could be explained if prophase pathway components are activated by these kinases *in vivo* which are added in their already active, phosphorylated form to our *in vitro* system. In this context it is interesting to note that Wapl was reported as a putatively Plk1-activated prophase pathway component (Challa, Fajish et al., 2019). Notably, we isolate Wapl from mitotically arrested cells, i.e. at a state of high Plk1 and Cdk1-cyclin B1 activity. Therefore, it will be interesting to test whether phosphatase treatment of Wapl may be sufficient to establish a Plk1 and/or Cdk1-cyclin B1 requirement for cohesin release in our assay. Finally, given their similarities, the actual *in vivo* requirement of Cdk1-cyclin B1 and/or Cdk1/2-cyclin A2 for prophase pathway signaling needs careful evaluation. While both kinases clearly have similar substrate specificities (Moore, Kirk et al., 2003), in our hands Cdk1/2-cyclin A2 can phosphorylate Pds5b, while Cdk1-cyclin B1 cannot (Figure 3A). Thus, it is conceivable that Cdk1/2-cyclin A2 can functionally replace Cdk1-cyclin B1 in prophase pathway signaling, but not *vice versa*."

...and also the reconstitution assay is not preformed in a quantitative manner. Because the Nek2a and Cdk1/2-cyclin A2 kinases are novel regulators in sister chromatid cohesion mechanism, more detailed descriptions of Nek2a- and Cdk1/2-cyclin A2-dependent phosphorylation would be needed.

Specific major concerns:

1) In the reconstitution experiments in the figure 1 and 2, protein purities and kinase activities are not well described. The authors should show Coomassie staining of purified proteins that they added (also in Fig EV4), and also describe their final concentrations. In addition, kinase

activities are not quantitatively tested. Though it is shown that all Nek2a, Cdk1/2-cyclin A2, and Cdk1-cyc B1 are somewhat active (Figure EV1A), it should be shown in more quantitative way (e.g. showing the time-course of substrate phosphorylation).

Since most, if not all, of cohesin is released from chromatin in our reconstitution experiments, Plk1 and/or Cdk1-cyclin B1 could only be required if they were present as contaminants in the preparations of Wapl, Nek2a, Cdk1/2-cyclin A2 and/or aurora B. Moreover, these contaminants would have to be more active than our actual preparations of Plk1 and Cdk1-cyclin B1. We can fully exclude this possibility for the following reasons:

- 1) We have used the same preparations of Plk1 and Cdk1-cyclin B1 in other projects to successfully phosphorylate substrates and therefore know that they are highly active.
- 2) The Aurora B-INCENP complex was isolated from bacteria and therefore cannot be a source of Plk1 and/or Cdk1-cyclin B1.
- 3) Since we isolate Wapl, Nek2a and Cyclin A2 from transfected Hek293T cells by single affinity chromatography, they are probably not very pure. However, Plk1 and cyclin B1 are not detectable by immunoblotting in these preparations (new Figure EV1B). Thus, we can only conclude that Plk1 and/or Cdk1-cyclin B1 are indeed dispensable for Wapl-dependent release under the conditions of our assay, although (and as discussed above) they may very well be required *in vivo*. Given these facts, we frankly don't see the need to determine the concentrations of all the kinases. We would still not know what fraction of each kinase is active and what fraction is inactive due to misfolding, lack of activating phosphorylations, presence of inhibitory phosphorylations, etc. Similarly, we would learn only little from time courses of model substrate phosphorylation. Such experiments will not tell us how much activity of a given kinase is required in cells for the effective phosphorylation of its physiologically relevant target. In other words, while we agree with the referee that it is not clear to what extent our reconstitution system recapitulates the physiological state, we don't think that the proposed experiments would change this shortcoming which is a general characteristic of most reconstitution experiments. Importantly, we did not solely rely on reconstitutions but confirmed the key observations of our biochemical assays by corresponding experiments in cells.

2) To see to what extent the reconstitution system sensitively recapitulates cellular events, it would be testable how much chromatin bound proteins are dissociated or stabilized after siRNAs for Sororin, Esco1/2, or Wapl. For instance, do cohesin subunits (Smc3, Rad21, or Stag1) still stay on chromatin after Sororin RNAi? If yes, are these cohesin complexes still resistant to kinase treatment without Wapl?

We prepared fresh G2 chromatin from mock-treated *versus* sororin-depleted cells and characterized it by release assay. The result is shown as new figure 8B and described at the end of the result section (page Z): "... Finally, to test whether phosphorylation might be sufficient to displace sororin-free cohesin from DNA, we also prepared chromatin beads from Sgo1-depleted G2 phase cells. Not surprisingly, this resulted in beads with a much reduced cohesin load. Nevertheless, Pds5a/b, Nipbl and Rad21 were detectable, did not elute upon incubation with Nek2a, Cdk1/2-cyclin A2 and aurora B-INCENP alone but still required Wapl for their release into the supernatant (Figure 8B, lanes 9-11)...."

3) In Figure EV3B, it is shown that pS1209 phosphorylation appears only in soluble fraction in released sample (lane 17). However, it is unclear when and where the phosphorylation by Nek2a and Cdk1/2-cycA2 occurs during cell cycle. The authors should describe this.

To follow the spatiotemporal dynamics of Ser1209-phosphorylation, we synchronously released HeLaK cells from a G2-arrest into mitosis. Every 5 minutes aliquots of cells were fractionated into chromatin and cytosol and analyzed by immunoblotting. The results of this experiment are shown as new figure 3D and described in the result section (page 5) as follows: "Twenty minutes after the release, chromatin-associated Pds5b was first found phosphorylated on Ser1209. Remarkably, this correlated perfectly with the recruitment of virtually all Wapl to chromosomes (Figure 3D, lanes 3 and 15). Five minutes later, most of Rad21, Pds5b, sororin and Wapl had already left the chromosomes (Figure 3D, lanes 4 and

16). Once cytosolic, Pds5b was slowly dephosphorylated, coinciding with the previous degradation of Nek2a and cyclin A2 (Figure 3D, lanes 4-7). Chemical inhibition of Nek2a at the time of release abolished Ser1209 phosphorylation and, importantly, delayed dissociation of cohesin from chromatin until separase activation (Figure 3D, lanes 9-12 and 21-24) (Lebraud et al., 2014). Thus, *in vivo* phosphorylation of Ser1209 occurs in early mitosis and in a Nek2a-dependent manner. Furthermore, this experiment demonstrates an *in vivo* requirement of Nek2a for prophase pathway signaling."

4) In Figure 2B (4B in the revised ms), the authors should include negative controls, where wild-type Pds5b were bound to the beads with or without Nek2a and Cdk1/2-cyclin A2. Though the result might be evident from previous results, if the Pds5b purification was done in the different way from different cells, then negative controls are always needed in each experiment.

Pds5b was always purified in the same manner, i.e. via Flag pull down from transfected, taxol arrested Hek293T cells. (We never detected co-purification of endogenous Wapl or sororin; see Figure 4A for an example of corresponding blank blots). Flag-Pds5b-WT, which has been purified in this manner, binds recombinant sororin but not Wapl when offered our mixture of both (Figure 4A, lane 1). We admit to have left out this "Figure 4A, lane 1"-like control in the experiment shown in figure 4B to reduce the density of the data and make the figure more readily accessible to the reader. However, we feel that this does not compromise the conclusion. We had previously repeated this (laborious) experiment to ensure its reproducibility (see below). Because the effort would be very high but we would not learn anything new, we refrained from repeating it yet again only to include this control.

5) In Figure 3B (5B in the revised ms), what is the consequence if both Nek2a Δ MR and Δ 86-cyclin A2 were expressed with Pds5-5A mutant?

We have not done this experiment. But given the mild phenotype caused by overexpression of Nek2a Δ MR, we would expect that the result would look much like column/lane 4, i.e. that the PCS caused by Δ 86-cyclin A2 would be suppressed.

6) In Figure 5A (7A in the revised ms), it is shown that Pds5b-5A binds to Sororin but not Wapl in prometaphase in whole cell extract. How about chromatin fraction? Is the association between Sororin and Pds5b-5A found on chromatin too?

As figure 7A shows, the more sororin is associated with Pds5b, the more histone H2A also co-purifies with Pds5b. We take this as a hint that the sororin-Pds5b interaction occurs primarily, if not exclusively, at chromatin.

Minor point:

7) In Figure 2B, headings of Pds5b's 2A, 3A and 5A are confusing. Please indicate 2A is from lane 6 to 8, 3A is from lane 9 to 11, and 5A is from lane 12 to 14.

We made this clearer in the revised manuscript

Referee #3:

In this manuscript Helmuth and Stemmann have provided a very careful analysis of the mechanisms that result in cohesin removal from chromosome arms during mitotic prophase. Here they focus on Pds5B, and the competition between Wapl and Sororin for Pds5B binding, and cohesin release in prophase. The data are thorough and offer important insight into how this mechanism is regulated in vertebrate models. It should be published.

We thank the referee for this positive overall assessment of our work.

I strongly recommend efforts to improve the readability.

Major concerns:

1. The writing is extremely dense and at times non-linear, and should be streamlined and possibly shortened, to make the manuscript more readable. For example, figure EV1 first shows kinase assays before telling us why they are being done. Better to introduce all kinases as possible modulators of the pathway, then walk through the data in EV1B (this would prevent going back and forth EV1A, EV1B, EV1A, EV1B, etc. making the writing more readable). More paragraph breaks to define each question/conclusion in the results section would also have been helpful. Some of the details in the results section should be moved to the figure legends, some of which are very minimal. Detailed EV panel lane callouts in the main text probably aren't necessary. Trust the reader! I don't know if the authors were panel-limited in the main figures, but it might help to move some of the EV data to the main figures.

We apologize for our shortcomings on the writing. In the revised manuscript, we have tried to heed all of the referee's advice to improve the readability.

2. A cartoon or table summarizing the various P-site mutants in the Pds5B tail, with the kinases thought to modify the residues (or color code the residues), and the mutant names would be great.

We have improved the corresponding cartoon to now also include the kinase information (Figure 3B of the revised ms).

3. A cartoon explaining their model, integrating what is known about Pds5A vs Pds5B, centromeres vs arms, would be very useful.

We have added the below model as part of the revised ms (new figure 1A):

Legend: Pericentromeric cohesin is deprotected long before sister chromatids separate. During phosphorylation- and Wapl-dependent release of cohesin from chromosome arms, pericentromeric cohesin is protected by associated Sgo1-PP2A. Following the early mitotic degradation of Nek2a and cyclin A2, Sgo1-PP2A relocates to kinetochores. Yet, pericentromeric cohesin persists until securin and cyclin B1 are degraded and separase is activated. A requirement for Nek2a and cyclin A2 for prophase pathway signaling would explain why in metaphase Wapl can no longer release cohesin.

4. Discussion: The discussion of Pds5A is important, and a bit confusing. The phosphorylation sites identified and characterized in Pds5B are not present in Pds5A, and many of their experiments are done in the absence of Pds5A. Their findings are consistent with previous observations in the literature (centromere vs arm cohesion), but it would have been helpful if they could discuss the relative roles of Pds5A and Pds5B and the impacts of their differences in regulation, if any, on mitotic chromosome structure and function. The discussion of Pds5A seems to be suggesting that Pds5A is similarly controlled by Nek2a, but not through phosphorylation of the same residues, or perhaps through modification of an additional protein.

We have prepared fresh 'click-DNA' (see next point) and conducted additional release assays. These experiments clearly show that, much like Pds5b, Pds5a is associated with G2 chromatin and evicted upon treatment with Nek2a, Cdk1/2-cyclin A2, aurora B-INCENP, Wapl and ATP (new Figure 8B). Thus, while Pds5a on its own is not directly affected by Nek2a and Cdk1/2-cyclin A2 in its binding behavior towards sororin and Wapl, Pds5a-cohesin is nevertheless susceptible to Wapl-dependent release. We cannot yet explain these observations but speculate in the discussion as follows (page 11): "Since Aurora B-dependent phosphorylations appear to be insufficient for cohesin release (Figure 1B), Nek2a and/or Cdk1/2-cyclin A2 are likely required also for opening of Pds5a-cohesin. This suggests the existence of an additional target of these kinases, with sororin being an obvious candidate. Alternatively, Rad21 and/or Smc3, both of which are phosphorylated by Nek2a in vitro (Figure EV2), may be relevant substrates. This is because their NHD and neck region, respectively, represent cohesin's exit gate which is predicted to also make contact with sororin (Nasmyth et al., 2023)."

Minor concerns (in no particular order):

1. Can they discuss inter-sister vs loop cohesion in their reconstitution system. Do they need to use G2 chromatin for the release assay to work? If so, why? Are they really just assaying inter-sister cohesion? Is this Pds5B-dependent mechanism uniquely involved in sister cohesion in prophase?

Our group is primarily interested in the regulation of sister chromatid cohesion rather than chromatin structure/transcription. Accordingly, the focus of this project was on cohesive cohesin and, in particular Pds5b-cohesin, because it maintains cohesion from prophase until

the onset of anaphase. Thus, we had to use G2 chromatin because we needed sister chromatids and passage through S-phase for the incorporation of the alkine-containing thymidine analog. However, G2-chromatin is expected to contain all three known types of cohesin. In addition to cohesion-mediating Pds5a/b-cohesin, which entraps two sister chromatids (in trans), it should also be loaded with the two types that act in cis: Nipbl-cohesin, which actively extruded DNA loops, and cohesin at CTCF-sites (TAD-boundaries), whose DNA 'pumping' activity has been arrested by swapping of Nipbl for Pds5a/b. In response to the reviewers' curiosity, we repeated the release assay and also followed the behavior of Nipbl and Pds5a (new figure 8B). This experiment showed that, similar to Pds5b, Pds5a and Nipbl are also released under the conditions of our assay. In the revised ms, we discuss these new findings in the light of our results for Pds5b-cohesin (page 11).

2. Introducing potential roles of Nek2 and cyclin A in the prophase pathway in the introduction would have been helpful.

Done in the revised manuscript.

1. Not sure what is meant by Cdk1/2 when you are talking about a specific reagent. Can you please explain? Was this cyclin IP with mixed kinases?

Exactly! Hek293T cells were transfected to express eGFP-SUMOstar-tagged, non-degradable cyclin A2 and then arrested in prometaphase with taxol. Cyclin A2 was affinity-purified from corresponding lysates with immobilized anti-eGFP nanobody and eluted together with associated kinases by incubation of the beads with SUMOstar protease.

2. First line in the Results section, should read from (says "form")

Corrected in the revised manuscript.

3. The use of "see below" is confusing. Please indicate exactly where this is shown, or change the order of the callouts in the text.

Changed to "see next paragraph" in the revised manuscript.

4. "Resistant against" should be "resistant to" (results page 2).

Corrected in the revised manuscript.

5. "Surprisingly, in the presence of Wapl, Nek2a, and Cdk1/2-cyclin A2, aurora B-INCENP was necessary and sufficient for removal of cohesin and sororin from chromatin..." This is a strange construction. How about "was able to ..., while Cdk/cyclin B was not"? Sufficiency is not really the appropriate term when you have wapl, nek2a and cdk/cyclin in the reaction.

Changed to: "In the presence of Wapl, Nek2a and Cdk1/2-cyclin A2, the addition of (bacterially expressed) aurora B-INCENP resulted in the release of cohesin and sororin from chromatin, whereas the addition of Cdk1-cyclin B1 or Plk1 had no effect (Figure 1C)"

6. Please provide a reference or description of the "stabilized truncation variants" of Nek2a and cyclin

In the revised manuscript, we now cite in this context the following two publications:

- den Elzen, N. & Pines, J. Cyclin A is destroyed in prometaphase and can delay chromosome alignment and anaphase. *J Cell Biol* 153, 121-136, doi:10.1083/jcb.153.1.121 (2001).

- Sedgwick, G. G. et al. Mechanisms controlling the temporal degradation of Nek2A and Kif18A by the APC/C-Cdc20 complex. *EMBO J* 32, 303-314, doi:10.1038/emboj.2012.335 (2013).

7. What are HeLaK cells?

The K stands for the Kyoto isolate of HeLa cells (Research Resource Identification: CVCL_1922; see, for example, *Cell* 163 (2015) 712-723; doi: 10.1016/j.cell.2015.09.053).

8. Numerous spelling errors, including argueing, agrevated. Please check thoroughly.

We apologize for this sloppiness and have checked the spelling more carefully in the revised manuscript.

9. How does this fit with the initial observation that Plk-dependent phosphorylation of SA2 causes cohesin removal (Hauf et al 2005)?

Under the conditions of our assay, Plk1 is dispensable for Wapl-dependent cohesin release. This could mean that *in vitro* Nek2a, Cdk1/2-cyclin A2 and/or aurora B-INCENP can functionally replace Plk1 in SA2 phosphorylation. Or it could mean that SA2 phosphorylation is not essential for opening of Pds5b-cohesin. In fact, the impairment of chromosome arm separation caused by expression of Plk1-resistant SA2-12A appears relatively mild in a more recent publication from the Peters group (PNAS 110 (2013) 13404-9; doi: 10.1073/pnas.1305020110). Moreover, the Yu-lab reported that SA2 in SGO1-bound cohesin is at least partially phosphorylated and questioned whether it is a substrate of PP2A (NCB 15 (2013) 40-9; doi: 10.1038/ncb2637).

An interaction with SA2 was reported for both sororin and Wapl (NSMB 21 (2014) 864–870; doi:10.1038/nsmb.2880; Cell Cycle 14 (2015) 820-6; doi: 10.1080/15384101.2014.1000206). Therefore, it may be possible that phosphorylation of sororin and/or Wapl impacts their binding to SA2 irrespective of or in addition to phosphorylation of SA2 itself.

By preparing immobilized G2-chromatin from cells, in which endogenous SA2 was replaced by SA2-12A, we will now be able to assess the importance of SA2 phosphorylation in our release assay. However, we feel that this is beyond the scope of this study. Because we did not yet do any experiments in this direction, we prefer not to speculate at this time about the importance of SA2 as a substrate of prophase pathway signaling.

+10. It would be nice to have a table for antibodies and reagents, instead of a long form paragraph

Done for all the antibodies! (Because we used only few reagents (small molecules) we refrained from preparing a table for them.)

Dr. Susanne Hellmuth
University of Bayreuth
Chair of Genetics
Universitaetsstrasse 30
Bayreuth 95440
Germany

9th Aug 2024

Re: EMBOJ-2024-117218R
Requirement of Nek2a and cyclin A2 for Wapl-dependent removal of cohesin from prophase chromatin

Dear Dr. Hellmuth,

Thank you for submitting your revised manuscript for our consideration, and my sincere apologies for its delayed re-evaluation, due to limited referee availability at this time. The original reviewers 2 and 3 have now looked at it once more, and are largely satisfied with your responses and revisions. After incorporation of minor textual modifications requested by referee 2, we shall therefore be happy to accept your manuscript for EMBO Journal publication.

In addition, there are also a few editorial issues that should still be addressed at this point:

- Importantly, please complete and provide the Source Data checklist sent to you by our Source Data Scientific Coordinator, Hannah Sonntag (I am attaching it once more to this email). Please note that the requests in this list still refer to the figure arrangement/labelling of the initial version, and make sure to explain how the currently provided Source Data files correspond to these.
- We note that the error bars are not defined in the legend of Figure 7b, please correct.
- Please note that Materials and Methods need to be described in the main text using our 'Structured Methods' format, including a Reagents and Tools Table (listing key reagents, experimental models, software and relevant equipment and including their sources and relevant identifiers) followed by a Methods and Protocols section. Please find a template (.docx) for the Reagents and Tools Table attached to this message. See our Guide to Authors for further information.
- Please rename the Conflict of Interest section into "Disclosure and Competing Interests Statement", in accordance with our updated Guide to Authors (<https://www.embopress.org/competing-interests>)
- As we are switching from a free-text author contribution statement towards a more formal statement based on Contributor Role Taxonomy (CRediT) terms, please remove the present Author Contribution section and instead specify each author's contribution(s) directly in the Author Information page of our submission system during upload of the final manuscript. See <https://casrai.org/credit/> for more information.
- Finally, please provide suggestions for a short 'blurb' text prefacing and summing up the conceptual aspect of the study in two sentences (max. 250 characters), followed by 3-5 one-sentence 'bullet points' with brief factual statements of key results of the paper; they will form the basis of an editor-written 'Synopsis' accompanying the online version of the article. Please also upload a synopsis image, which can be used as a "visual title" for the synopsis section of your paper. The image should be in PNG or JPG format, and please make sure that it remains in the modest dimensions of (EXACTLY) 550 PIXELS WIDE and 300-600 PIXELS HIGH.

I am therefore returning the manuscript to you for a final round of revision, to allow you to make these modifications and upload the revised files. Once we will have received them, we should be ready to swiftly proceed with formal acceptance and production of the manuscript.

Yours sincerely,

Hartmut Vodermaier

*** PLEASE NOTE: All revised manuscripts are subject to initial checks for completeness and adherence to our formatting guidelines. Revisions may be returned to the authors and delayed in their editorial re-evaluation if they fail to comply to the following requirements (see also our Guide to Authors for further information):

9) To facilitate reproducibility and cross-laboratory adoption of methodologies, please structure the Materials & Methods section as outlined in our guide to authors, including a completed Reagents and Tools Table that can be downloaded from our author guidelines as well (<https://www.embopress.org/page/journal/14602075/authorguide#structuredmethods>).

10) Digital image enhancement is acceptable practice, as long as it accurately represents the original data and conforms to community standards. If a figure has been subjected to significant electronic manipulation, this must be clearly noted in the figure legend and/or the 'Materials and Methods' section. The editors reserve the right to request original versions of figures and the original images that were used to assemble the figure. Finally, we generally encourage uploading of numerical as well as gel/blot image source data; for details see: embopress.org/page/journal/14602075/authorguide#sourcedata

At EMBO Press, we ask authors to provide source data for the main manuscript figures. Our source data coordinator will contact you to discuss which figure panels we would need source data for and will also provide you with helpful tips on how to upload and organize the files.

In the interest of ensuring the conceptual advance provided by the work, we recommend submitting a revision within 3 months (7th Nov 2024). Please discuss the revision progress ahead of this time with the editor if you require more time to complete the revisions. Use the link below to submit your revision:

Link Not Available

Referee #2:

I re-read the revised manuscript by Hellmuth and Stemmann. The authors answered most of the points, including clear description and discussion about dispensability of Plk1 and Cdk1-cycB, sensitivity of their Dm-ChP method for Sororin depletion (Figure 8B), and new time-course experiment of pS1209 phosphorylation timing and the place (Figure 3D). It is a bit surprising to see all Wapl molecules seem to be bound to chromatin very transiently 20 min after G2 release. This result well fits the authors model. However, as the original Wapl paper (Kueng et al 2006 Cell) showed that Wapl could be bound to interphase chromatin in cohesin-dependent manner, undetectable level of Wapl in G2 chromatin pellet at 0 min (Figure 3D) looks inconsistent with previous observations. It could be because of the cell type (HeLa K cell) or other critical differences from previous assay system. It could be better to mention this point in text. After improvement in this point, I would recommend this study for the publication.

Minor point: Figure legend of 1C "as shown in B"?

Referee #3:

The authors have done an excellent job addressing all of the reviewers' concerns. I'm particularly grateful for the added cartoons and models. I am happy to accept as is currently written.

All editorial and formatting issues were resolved by the authors.

Dr. Susanne Hellmuth
University of Bayreuth
Chair of Genetics
Universitaetsstrasse 30
Bayreuth 95440
Germany

27th Aug 2024

Re: EMBOJ-2024-117218R1

Requirement of Nek2a and cyclin A2 for Wapl-dependent removal of cohesin from prophase chromatin

Dear Dr. Hellmuth,

Thank you for submitting your final revised manuscript for our consideration. I am pleased to inform you that we have now accepted it for publication in The EMBO Journal.

Yours sincerely,

Hartmut Vodermaier
